# Solving Spatial-Spectral Fusion with Latent Spectral Operators

**Wei Li** [1]   **Jieyuan Pei** [1]   **Junnan Xu** [1]   **Xuanfeng Ding** [1]   **Junwei Zhu** [1]   **Wanjun Chen** [1]   **Jianwei Zheng** [* 1 2]

**Project Page:** https://weili419.github.io/latent-spectral-operators

## Abstract

Existing deep spatial–spectral fusion (SSF) methods typically learn the fusion mapping in the coordinate domain using convolutions and attentions, making it hard to scale across varying spatial resolutions and offering limited control over the frequency content of the reconstructions, which may further lead to severe spectral distortion. In this work, we propose Latent Spectral Operators (LSO), a SSF framework that learns fusion mappings between spectral functions through a structured operator parameterization. Specifically, LSO first applies a cross-attention projection, where learned latent tokens serve as spectral prompts, to compress high-dimensional observations into a compact latent representation, and then adopts a hierarchical, patch-based architecture to integrate rich multi-scale cues. Furthermore, to parameterize the latent fusion operator in a controllable manner, a Trigonometric Basis Solver is elaborated, which represents the mapping using a trigonometric basis expansion. This formulation naturally supports multi-frequency modeling, with a capacity–stability trade-off governed by the number of basis functions. Extensive experiments on the CAVE and Harvard benchmarks demonstrate that LSO achieves consistent state-of-the-art performance and exhibits strong transferability across different spatial scales.

## 1. Introduction

Hyperspectral images (HSIs) capture hundreds of narrow, contiguous spectral bands spanning the visible to near-infrared wavelengths, enabling a wide range of applications such as change detection (Feng et al., 2025b;a), disaster assessment (Guo et al., 2025), and geolocation (Song et al., 2025). Compared with conventional RGB imagery, HSIs provide a high-dimensional spectral signature at each pixel, encoding not only spatial appearance but also material-dependent spectral characteristics (Liu et al., 2025c). Despite their rich spectral content, hyperspectral imaging is fundamentally constrained by the trade-off between spectral and spatial resolution (Zhang et al., 2024): acquiring more bands typically reduces spatial resolution, whereas enhancing spatial size often requires sacrificing spectral information. Consequently, practical systems often capture the same scene with complementary sensors, yielding a high-spatial-resolution multispectral image (HR-MSI) and a low-spatial-resolution hyperspectral image (LR-HSI). Spatial–Spectral Fusion (SSF) (Ma et al., 2025) leverages this complementarity to reconstruct a high-spatial-resolution hyperspectral image (HR-HSI). Existing SSF methods generally fall into two categories: prior-based approaches (Xu et al., 2025a) and deep learning-based approaches (Sun et al., 2023; Cao et al., 2024). Prior-based methods rely on hand-crafted regularization and explicit constraints, but often struggle to recover high-fidelity HR-HSIs, especially with limited training data. Deep learning methods instead learn the complex fusion mapping from data, typically combining convolutional architectures (Ran et al., 2024) for strong local inductive biases with attention mechanisms (Ma et al., 2024; Jia et al., 2023) to model long-range dependencies. Moreover, SSF benefits from multi-scale reasoning (Huang et al., 2024; Liang et al., 2025), since both fine textures/edges and large-scale structures are essential for faithful spatial detail reconstruction. However, most existing methods often learn a resolution-tied mapping in the pixel/coordinate domain, which can generalize poorly when the input resolutions of HR-MSI and LR-HSI vary (Liu et al., 2025b; Zhu et al., 2025; Jiang et al., 2026). In this paper, we aim to decouple the coordinate (pixel) space from the learned features, thereby alleviating this limitation.

In recent years, latent-space-based algorithms (Qin et al., 2025) have gained increasing attention, largely because they shift learning from the complex, high-dimensional pixel domain to a more compact latent representation. Such approaches have been extensively explored in natural im-

[1]College of Computer Science and Technology, Zhejiang University of Technology, Hangzhou 310023 [2]Zhejiang Key Laboratory of Visual Information Intelligent Processing, Hangzhou 310023. Correspondence to: Jianwei Zheng <zjw@zjut.edu.cn>.

*Proceedings of the 43rd International Conference on Machine Learning*, Seoul, South Korea. PMLR 306, 2026. Copyright 2026 by the author(s).

age modeling (Kouzelis et al., 2025), large language models (Liu et al., 2025a), and partial differential equation solving (Wang & Wang, 2024), but remain largely underexplored for SSF. Motivated by these advances, we introduce a cross-attention-based projection module (Cui et al., 2025) that embeds HSIs into a compact latent space, thereby improving scalability as spatial resolution increases. With the projection in place, the key remaining question becomes *how to parameterize the fusion mapping in latent space*. A straightforward choice is to adopt generic backbones, such as convolutional (Raonic et al., 2023), attention-based (Su et al., 2025), or Mamba-style (He et al., 2025) architectures. While powerful, these black-box parameterizations provide limited explicit control over which frequency components are amplified. This limitation is particularly problematic for SSF: the target HR-HSI should remain smooth along the spectral dimension (across wavelengths) while recovering sharp, high-frequency details in the spatial domain (edges/textures). Without structured control, high-frequency energy may inadvertently leak into the spectral dimension, causing severe distortion. To address this issue, we propose a Trigonometric Basis Solver, which represents the latent fusion operator as a trigonometric basis expansion. This structured parameterization naturally captures multi-frequency components (Li et al., 2025a) and provides a simple control knob, namely the number of basis functions, to balance expressivity and stability.

Based on this insight, we propose **Latent Spectral Operators (LSO)**, which learn fusion mappings between spectral functions in latent space via a trigonometric basis expansion. Specifically, we design a cross-attention projection network in which learned latent tokens act as spectral prompts to compress high-dimensional observations into a compact latent representation. This projection emphasizes physically meaningful spectral characteristics while avoiding resolution-tied coordinate representations. In addition, we adopt a hierarchical patch-based architecture to inject rich multi-scale cues that are critical for SSF. Experimentally, LSO achieves consistent state-of-the-art performance on two well-established benchmarks and shows strong transferability across HSIs with varying spatial scales. Our contributions are summarized as follows.

- We propose Latent Spectral Operators, a new SSF framework featuring a cross-attention projection with learned latent tokens (spectral prompts) and a hierarchical patch-based design to effectively incorporate multi-scale cues under varying spatial resolutions.

- A Trigonometric Basis Solver is elaborated to parameterize the latent fusion operator via a structured basis expansion, enabling multi-frequency modeling and an explicit capacity–stability trade-off, supported by theoretical analysis.

- Extensive experiments on CAVE and Harvard demonstrate state-of-the-art reconstruction accuracy and strong cross-scale transferability.

## 2. Related Works

### 2.1. Spatial-Spectral Fusion

Traditionally, SSF has been addressed by exploiting handcrafted priors, such as low-rankness (Liu et al., 2025c) and self-similarity (Hu et al., 2024). These prior-based methods are often interpretable, yet their handcrafted constraints may fail to capture real data statistics, resulting in limited reconstruction fidelity. Moreover, they typically rely on computationally expensive optimization and require careful model selection, which hinders practical deployment.

With the success of deep learning, numerous CNN-based (Ran et al., 2024; Zhang et al., 2020; Li et al., 2025c) and Transformer-based (Yang et al., 2023; Su et al., 2025; Li et al., 2026; Zhu et al., 2026; Xu et al., 2026) SSF methods have been proposed, benefiting from end-to-end feature learning for high-quality fusion. Hybrid CNN–Transformer designs further combine local inductive biases with global dependency modeling, and have become a prevailing choice (Hou et al., 2024). More recently, to improve generalization across spatial resolutions, researchers have introduced neural operators (Liu et al., 2025b) and implicit neural representations (INRs) (Liang et al., 2024; Xu et al., 2025b) into SSF, as both aim to learn mappings between continuous functions. For instance, Deng et al. (Deng et al., 2025) combine an octree structure with INR to integrate spatial and spectral information. Nevertheless, most of these approaches still parameterize fusion mapping in coordinate domain, which ties the model to a specific spatial resolution. Moreover, they provide limited explicit control over how high-frequency spatial details are injected, and such uncontrolled injection may leak into spectral dimension and cause distortion.

### 2.2. Latent Space Learning

Latent-space-based algorithms shift learning from the high-dimensional pixel/coordinate domain to a compact hidden representation, and have been widely explored in natural image modeling (Kouzelis et al., 2025; Li et al., 2025b; Gu et al., 2025b;a; Zheng et al., 2025), large language models (Liu et al., 2025a), and partial differential equation solving (Wang & Wang, 2024; Wu et al., 2023; Zheng et al., 2024; Lu et al., 2026; Chen et al., 2026). Within this line of work, a common strategy is to learn an explicit transformation between the coordinate and latent domains. For example, Liu et al. (Liu et al., 2025b) use pointwise convolutions for this transformation, while Wu et al. (Wu et al., 2025) adopt a stacked convolutional encoder. Different from

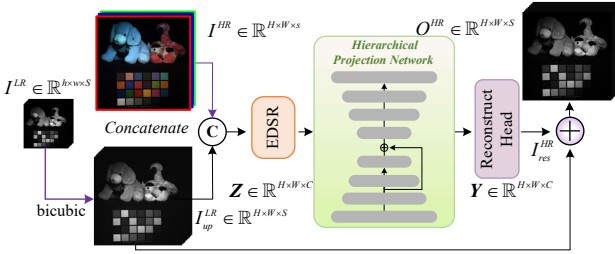

*Figure 1.* Overall design of Latent Spectral Operator.

these convolution-based projections, we introduce learnable latent tokens and use cross-attention to project hyperspectral images into the latent space, enabling fusion mappings to be learned directly in the latent domain.

# 3. Latent Spectral Operator

As discussed, high-dimensional SSF are challenging due to their prohibitive computational cost and the complexity of the underlying input–output mappings. To address these issues, we propose Latent Spectral Operator (LSO), featuring a *Hierarchical Projection Network* that compresses high-dimensional data into a compact space for improved efficiency. Moreover, inspired by spectral methods, we elaborate a *Trigonometric Basis Solver* that models intricate mappings using a collection of basis operators, and comes with strong approximation and convergence guarantees.

**Overall framework.** Fig. 1 provides an overview of our framework, which fuses LR-HSI $I^{LR} \in \mathbb{R}^{h \times w \times S}$ and HR-MSI $I^{HR} \in \mathbb{R}^{H \times W \times s}$ to generate HR-HSI $O^{HR} \in \mathbb{R}^{H \times W \times S}$ under an upsampling factor $r$. First, the bicubically upsampled LR-HSI $I_{up}^{LR} \in \mathbb{R}^{H \times W \times S}$ is concatenated with $I^{HR}$ and fed into an encoder (EDSR) to extract deep features $\mathbf{Z} \in \mathbb{R}^{H \times W \times C}$. These features are then passed to the Hierarchical Projection Network, producing the fused representation $\mathbf{Y} \in \mathbb{R}^{H \times W \times C}$. Next, $\mathbf{Y}$ is processed by a conv-based module to adjust the channel dimension, yielding a residual image $I_{res}^{HR} \in \mathbb{R}^{H \times W \times S}$. Finally, the residual $I_{res}^{HR}$ is added element-wise to the bicubically upsampled image $I_{up}^{LR}$, producing the final fused output $O^{HR}$.

## 3.1. Encoder Network

Following the common design in neural networks for visual inverse problems (Wei & Zhang, 2023; Liang et al., 2024), the first stage of our LSO framework is to extract a latent code representation. We adopt EDSR (Lim et al., 2017) as the encoder, as it can produce latent features $\mathbf{Z} \in \mathbb{R}^{H \times W \times C}$ that preserve spectral information while enhancing spatial details. The encoding process is formulated as

$$\mathbf{Z} = \text{EDSR}_\psi\big(\text{Cat}\big(I_{up}^{\text{LR}}, I^{\text{HR}}\big)\big), \tag{1}$$

where $\text{EDSR}_\psi$ denotes the encoder parameterized by $\psi$, and $\text{Cat}(\cdot)$ represents channel-wise concatenation.

## 3.2. Hierarchical Projection Network

To avoid quadratic interactions in the dense coordinate space during spatial-spectral fusion, we propose a hierarchical projection network that integrates *latent-token cross-attention projectors* into a *hierarchical patch-based architecture*.

**Latent-token Cross-attention Projectors.** By projecting coordinate-wise representations into a small set of shared latent tokens and decoding them back, the network creates a compact latent bottleneck that enables scalable and effective fusion. Concretely, we introduce attention-based projectors equipped with latent tokens that are shared across all input–output pairs. These tokens are initialized as learnable parameters and optimized to capture dataset-wide commonalities—particularly the characteristic spectral responses. In this way, they serve as spectral-correlation prompts that guide the projection process.

Specifically, given coordinate set $\Omega$ containing $H \times W$ points and deep representations of inputs $\{\mathbf{Z}(x) \in \mathbb{R}^C\}_{x \in \Omega}$, we randomly initialize $N$ latent tokens $\{\mathbf{T}_i \in \mathbb{R}^{d_{\text{latent}}}\}_{i=1}^N$ to provide spectral-correlation prompts. As given in Fig. 2, a cross-attention projector is employed that uses latent tokens as queries and coordinate-wise features as keys and values, thereby aggregating information from the full spatial field into a compact latent representation. A residual connection is further incorporated to facilitate stable optimization. This process can be formalized as:

$$\mathbf{T}_{\mathbf{Z},i} = \mathbf{T}_i + \sum_{x \in \Omega} \frac{\text{Sim}(\mathbf{T}_i, \mathbf{Z}(x)\mathbf{W}_{\mathbf{k}})}{\sum_{x' \in \Omega} \text{Sim}(\mathbf{T}_i, \mathbf{Z}(x')\mathbf{W}_{\mathbf{k}})} (\mathbf{Z}(x)\mathbf{W}_{\mathbf{v}}), \tag{2}$$

where $i \in \{1, 2, \cdots, N\}$ and $\mathbf{W}_{\mathbf{k}}, \mathbf{W}_{\mathbf{v}} \in \mathbb{R}^{C \times d_{\text{latent}}}$ are linear layers for keys and values. $\text{Sim}(\mathbf{T}_i, \mathbf{Z}(x)\mathbf{W}_{\mathbf{k}}) = \exp\left(\mathbf{T}_i (\mathbf{Z}(x)\mathbf{W}_{\mathbf{k}})^\top / \sqrt{|\Omega|}\right)$ is for the similarity calculation. Under the spectral-correlation prompts of learned latent tokens $\{\mathbf{T}_i\}_{i=1}^N$, the deep representations $\{\mathbf{Z}(x)\}_{x \in \Omega}$ in the high-dimensional coordinate space are projected to $N$ tokens $\{\mathbf{T}_{\mathbf{Z},i}\}_{i=1}^N$ in latent space, where the latter is free from redundant coordinate information. To simplify notations, we summarize Eq. (2) as $\{\mathbf{T}_{\mathbf{Z},i}\}_{i=1}^N = \text{CoordToLatent}\big(\{\mathbf{T}_i\}_{i=1}^N, \{\mathbf{Z}(x)\}_{x \in \Omega}\big)$. After fusing spatial and spectral feature information in latent space with the *Trigonometric Basis Solver* (3.3), the latent input tokens $\{\mathbf{T}_{\mathbf{Z},i}\}_{i=1}^N$ are mapped to the latent output tokens $\{\mathbf{T}_{\mathbf{Y},i}\}_{i=1}^N$. We summarize the solving process in latent space as $\{\mathbf{T}_{\mathbf{Y},i}\}_{i=1}^N = \text{Solve}\big(\{\mathbf{T}_{\mathbf{Z},i}\}_{i=1}^N\big)$.

Finally, we need to project latent output tokens back to high-dimensional coordinate space as the final output. Similar to Eq. (2), by taking input representations as queries to provide coordinate information and latent output tokens as keys and

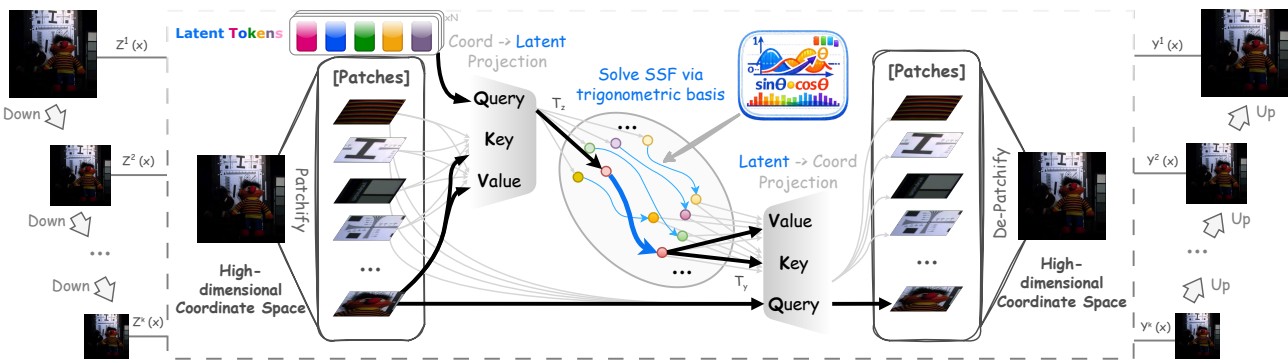

*Figure 2.* Overall design of the Hierarchical Projection Network.

values, this process can be formalized as follows:

$$Y(x)=Z(x)+\sum_{i=1}^{N}\frac{\text{Sim}(Z(x),T_{Y,i}W_k')}{\sum_{i'=1}^{N}\text{Sim}(Z(x),T_{Y,i'}W_k')}(T_{Y,i}W_v'),\tag{3}$$

where $x \in \Omega$ and $W_k', W_v' \in \mathbb{R}^{d_{\text{latent}} \times C}$ are linear layers for keys and values. Eq. (3) is summarized as $\{Y(x)\}_{x \in \Omega} = \text{LatentToCoord}\big(\{Z(x)\}_{x \in \Omega}, \{T_{Y,i}\}_{i=1}^{N}\big)$.

**Hierarchical Patch-based Architecture.** Notably, ground objects in remote sensing images exhibit substantial scale variation, and the scenes are complex and diverse. To better capture these inherent multiscale characteristics and the complex interactions in SSF, we propose a hierarchical patch-based architecture that performs SSF across regions at multiple spatial scales.

As illustrated in Fig. 2, given the raw encoder outputs $\{Z(x)\}_{x \in \Omega}$, we employ a parameterized downsampling layer to generate multi-scale deep representations $\big\{\{Z^k(x)\}_{x \in \Omega^k}\big\}_{k=1}^{K}$ across $K$ scales by aggregating local observations with learnable parameters. The finest-scale representation is $\{Z^1(x)\}_{x \in \Omega^1}$. At the $k$-th scale, we further perform a patchification operation (Deng et al., 2023) that partitions the coordinate set $\Omega^k$ into $P_k$ nonoverlapping patches $\{\Omega_j^k\}_{j=1}^{P_k}$ for region-wise processing. Here, $\Omega_j^k \subset \Omega^k$ denotes the coordinate subset of the $j$-th patch. Details of the downsampling and patchification operations are provided in Appendix A. Following the strategy in the preceding section, we randomly initialize a set of latent tokens $\{\{T_i^k\}_{i=1}^{N}\}_{k=1}^{K}$ at $K$ scales. The solving process for the $j$-th patch at the $k$-th scale is formulated as:

$$T_{Z,j}^k=\text{CoordToLatent}\Big(T^k, \{Z^k(x)\}_{x \in \Omega_j^k}\Big),$$
$$T_{Y,j}^k=\text{Solve}\big(T_{Z,j}^k\big),$$
$$\{Y^k(x)\}_{x \in \Omega_j^k}=\text{LatentToCoord}\Big(\{Z^k(x)\}_{x \in \Omega_j^k}, T_{Y,j}^k\Big),\tag{4}$$

where $T^k=\{T_i^k\}_{i=1}^{N}$. More details of $\text{Solve}(\cdot)$ are deferred into Subsection 3.3. Note that the patches at the same scale

come from the same underlying remote sensing images, whereas across scales the spatial texture details of the observations vary. Thus, the model parameters, e.g. latent tokens and linear layers, are shared among patches within the same scale but kept independent across different scales.

After the de-patchify operation, patches are spliced into the output at the $k$-th scale as $\{Y^k(x)\}_{x \in \Omega^k}$. We then successively upsample multi-scale outputs from coarse to fine. Concretely, at the $k$-th scale, $\{Y^k(x)\}_{x \in \Omega^k}$ is concatenated with the interpolation-upsampled $\{\text{Up}(Y^{k+1})(x)\}_{x \in \Omega^k}$. Finally, we obtain the finest-scale representations $\{Y^1(x) \in \mathbb{R}^C\}_{x \in \Omega}$, which are mapped to the final outputs $I_{res}^{HR} \in \mathbb{R}^{H \times W \times S}$ via a linear mapping parameterized in $\mathbb{R}^{C \times S}$.

### 3.3. Trigonometric Basis Solver

Benefitting from the hierarchical projection network, we address SSF by approximating the complex mapping between latent input-output tokens as described in Eq. (4). Inspired by classical Fourier-type expansions (Li et al., 2025a), we parameterize the solver as a linear combination of fixed trigonometric basis operators:

$$\mathcal{F}_{\text{Solve}}(\cdot) = \sum_{i=1}^{M} w_i \mathcal{F}_{\text{Solve},i}(\cdot),\tag{5}$$

where $M$ is an even hyperparameter, $\{\mathcal{F}_{\text{Solve},i}\}_{i=1}^{M}$ are fixed basis operators, and $\{w_i\}_{i=1}^{M}$ are learnable combination coefficients. Given latent input tokens $T_Z \in \mathbb{R}^{N \times d_{\text{latent}}}$ and let $z_i := T_Z(i) \in \mathbb{R}^{d_{\text{latent}}}$ denote the $i$-th token, we define $Q = \frac{M}{2}$ and construct the $M$ basis operators such as:

$$\mathcal{F}_{\text{Solve},(2q-1)}(z_i) = \sin(q\,z_i),$$
$$\mathcal{F}_{\text{Solve},(2q)}(z_i) = \cos(q\,z_i), \quad q \in \{1, \ldots, Q\},\tag{6}$$

where $\sin(\cdot)$ and $\cos(\cdot)$ act element-wisely on vectors. Technically, the trigonometric basis solver maps each token $z_i$ to an output token $y_i := T_Y(i) \in \mathbb{R}^{d_{\text{latent}}}$ via a residual form:

$$y_i = z_i + \gamma + \sum_{q=1}^{Q} \alpha_q \sin(q\,z_i) + \sum_{q=1}^{Q} \beta_q \cos(q\,z_i),\tag{7}$$

where $\boldsymbol{\gamma} \in \mathbb{R}^{d_{\text{latent}}}$ is a learnable bias term, and $\boldsymbol{\alpha_q}, \boldsymbol{\beta_q} \in \mathbb{R}^{d_{\text{latent}}}$ are learnable coefficients. Applying Eq. (7) to all tokens yields $\boldsymbol{T_Y} \in \mathbb{R}^{N \times d_{\text{latent}}}$, which we summarize as $\boldsymbol{T_Y} = \text{Solve}(\boldsymbol{T_Z})$. Similar to latent tokens, the parameters $\{\boldsymbol{\gamma}, \boldsymbol{\alpha}, \boldsymbol{\beta}\}$ are shared across patches within the same scale but are independent across different scales. Since the training pairs are generated under the same spectral response function governed image formation process, the model parameters are optimized to produce latent-space predictions that are consistent with the underlying response.

In summary, the overall fusion pipeline can be illustrated as the following commutative diagram:

$$
\begin{array}{ccc}
\boldsymbol{T_Z} & \xrightarrow{\quad \text{Solve} \quad} & \boldsymbol{T_Y} \\
{\scriptstyle \text{CoordToLatent}} \uparrow & & \downarrow {\scriptstyle \text{LatentToCoord}} \\
\mathcal{Z}(\hat{\Omega}; \mathbb{R}^C) & \xrightarrow{\text{Mapping}} & \mathcal{Y}(\hat{\Omega}; \mathbb{R}^C)
\end{array}
$$

Here, $\mathcal{Z}(\hat{\Omega}; \mathbb{R}^C)$ and $\mathcal{Y}(\hat{\Omega}; \mathbb{R}^C)$ denote the continuous low-dimensional manifolds on which the spectral features $\boldsymbol{Z}$ and $\boldsymbol{Y}$ reside, respectively, with $\hat{\Omega}$ representing the underlying continuous domain. Rather than learning the continuous manifold-to-manifold mapping directly in the coordinate space, we lift the fields to a latent token space via $\text{CoordToLatent}$ and learn the transformation using a trigonometric basis decomposition in the latent space (implemented by $\text{Solve}$), after which the result is projected back to the coordinate-space representation through $\text{LatentToCoord}$. In this way, the challenging continuous mapping on the manifolds is replaced by a structured and token-wise tractable approximation in the latent space.

*Remark* 3.1 (Why not approximate on the manifold/coordinate space directly). From the field-to-field viewpoint in the diagram, one may consider approximating the continuous mapping directly by trigonometric (Fourier-type) expansions on the underlying domain (or on the associated manifold charts) (Katznelson, 2004). However, once the fields are discretized on a grid $\Omega = \{x_j\}_{j=1}^{u}$, the induced map acts on the sampled values, i.e., $\tilde{g} : \mathbb{R}^{u \times C} \to \mathbb{R}^{u \times C}$. A naive spectral approximation of $\tilde{g}$ would amount to fitting multivariate trigonometric polynomials whose effective input dimension scales with the number of samples $u$ (one degree of freedom per location), rather than the intrinsic dimension of the underlying manifold. Consequently, truncating the corresponding Fourier series at frequency $J$ requires $(2J + 1)^u$ modes (Grafakos et al., 2008), and approximation rates typically deteriorate as $u$ grows (DeVore et al., 1994; Dyachenko, 1995). This makes direct spectral approximation in the discretized coordinate/manifold space prohibitively expensive and increasingly unstable as the spatial resolution increases.

*Remark* 3.2 (Solve reduces to a family of 1D function approximations). After lifting the coordinate-space fields to the latent tokens $\boldsymbol{T_Z}$ and applying a proper normalization so that each *channel coordinate* lies in $[0, \pi]$, the $\text{Solve}$ block is implemented by a finite trigonometric expansion with a residual connection. As a result, the transformation is *token-wise* and *entry-wise*: for any token index $i$ and channel coordinate $r$, the output value $(\boldsymbol{T_Y})_{i,r}$ depends only on the scalar input $(\boldsymbol{T_Z})_{i,r}$ through a one-dimensional mapping. Equivalently, learning $\text{Solve}$ amounts to approximating a family of scalar functions $f_r : [0, \pi] \to \mathbb{R}$ for $r = 1, \ldots, d_{\text{latent}}$, rather than approximating a multivariate function whose effective input dimension scales with the number of samples $u = |\Omega|$ in the discretized coordinate/manifold space (Remark 3.1). This separable one-dimensional structure enables tractable and scalable spectral approximation in the latent space. The claim is formalized in Lemma B.7, and the approximation rate under Lipschitz targets is given in Theorem B.8 and Corollary B.9.

## 4. Experiments and Analysis

### 4.1. Experimental Setting

**Datasets.** We evaluate our model on the CAVE and Harvard datasets. CAVE contains 32 hyperspectral images (HSIs), each with 31 spectral bands spanning 400–700 nm at a 10 nm interval. Following prior work (Deng et al., 2025), we use 21 images for training and the remaining 11 for testing. The Harvard dataset includes 77 HSIs covering both indoor and outdoor scenes, each with a spatial resolution of $1392 \times 1040$ and 31 bands in the 420–720 nm range. We select 20 images from Harvard, crop the upper-left region to $1000 \times 1000$, and split them evenly into 10 training images and 10 testing images.

Our model takes a pair of LR-HSI and HR-MSI observations, denoted as $(I^{LR}, I^{HR})$, and produces an HR-HSI output $O^{HR}$ for supervision. Since real paired measurements with ground-truth HR-HSIs are unavailable, we adopt a standard simulation protocol. For CAVE, we crop the 21 training images into 3,920 overlapping patches of size $64 \times 64 \times 31$, which serve as ground truth. To synthesize the corresponding LR-HSIs, we first blur each HR-HSI patch using a Gaussian kernel, and then downsample it by a factor of 4 or 8. To generate the HR-MSIs, we apply the Nikon D7003 spectral response function to the HR-HSI patches. This results in 3,920 LR-HSI patches of size $16 \times 16 \times 31$ and 3,920 HR-MSI patches of size $64 \times 64 \times 3$. The simulated samples and their ground-truth patches are randomly split into 80% for training and 20% for testing. We follow the same procedure for the Harvard dataset.

**Implementation Details and Quality Metrics.** We implement the proposed network in PyTorch 2.8.0 with Python 3.12. All experiments are conducted on a Linux workstation equipped with an NVIDIA H100 GPU (80 GB). We train

*Table 1.* The average and standard deviation calculated for all the methods on the CAVE dataset simulating scaling factors of 4 and 8. The best results are in bold, second-best in underline. Params and FLOPs are reported in millions (M) and billions (G), respectively. ∗ indicates $p < 0.01$, and † indicates $p < 0.05$.

| Methods | Params | FLOPs | CAVE ×4 | | | CAVE ×8 | | |
|---|---|---|---|---|---|---|---|---|
| | | | PSNR ↑ | SAM ↓ | ERGAS ↓ | PSNR ↑ | SAM ↓ | ERGAS ↓ |
| Bicubic | - | - | 31.175±3.428 | 4.683±1.692 | 9.629±5.020 | 27.383±3.287 | 6.449±2.433 | 14.349±6.179 |
| DCT$_{2024}$ | 8.15 | 2457.43 | 50.196±3.359* | 2.486±0.799* | 1.198±0.738† | 47.441±4.532* | 3.164±0.824* | 1.791±1.360* |
| LFormer$_{2024}$ | 2.28 | 306.94 | 49.847±3.712* | 2.584±0.746* | 1.313±0.890* | 47.302±4.389* | 3.474±1.046* | 1.813±1.329* |
| MIMO$_{2024}$ | 4.98 | **49.08** | 50.101±2.997* | 2.767±0.875* | 1.237±0.701* | 47.442±3.781* | 3.985±1.356* | 1.758±1.120* |
| FeINFN$_{2024}$ | 3.17 | 382.72 | 50.358±3.547* | 2.537±0.791* | 1.210±0.816* | 47.476±4.364* | 3.565±1.157* | 1.755±1.350* |
| KNLConv$_{2024}$ | **1.73** | 114.04 | 48.570±4.809* | 2.790±0.903* | 1.557±1.006* | 46.150±4.317* | 3.639±1.161* | 1.989±1.318* |
| SpecSolver$_{2025}$ | 3.10 | 364.75 | 50.568±3.895* | 2.489±0.793* | 1.212±0.935* | 47.831±4.899* | 3.527±1.152* | 1.805±1.698* |
| Otias$_{2025}$ | 2.99 | 278.35 | 50.213±3.438* | 2.538±0.784* | 1.211±0.799* | 47.769±4.272* | 3.201±0.921* | 1.757±1.324* |
| Ours | 1.94 | 222.85 | **50.728±3.575** | **2.381±0.728** | **1.170±0.804** | **48.118±4.589** | **2.961±0.838** | **1.675±1.427** |
| Ours vs. Best compe. | | | ▲0.160 (0.3%) | ▼0.105 (4.4%) | ▼0.028 (2.4%) | ▲0.287 (0.6%) | ▼0.203 (6.9%) | ▼0.080 (4.8%) |

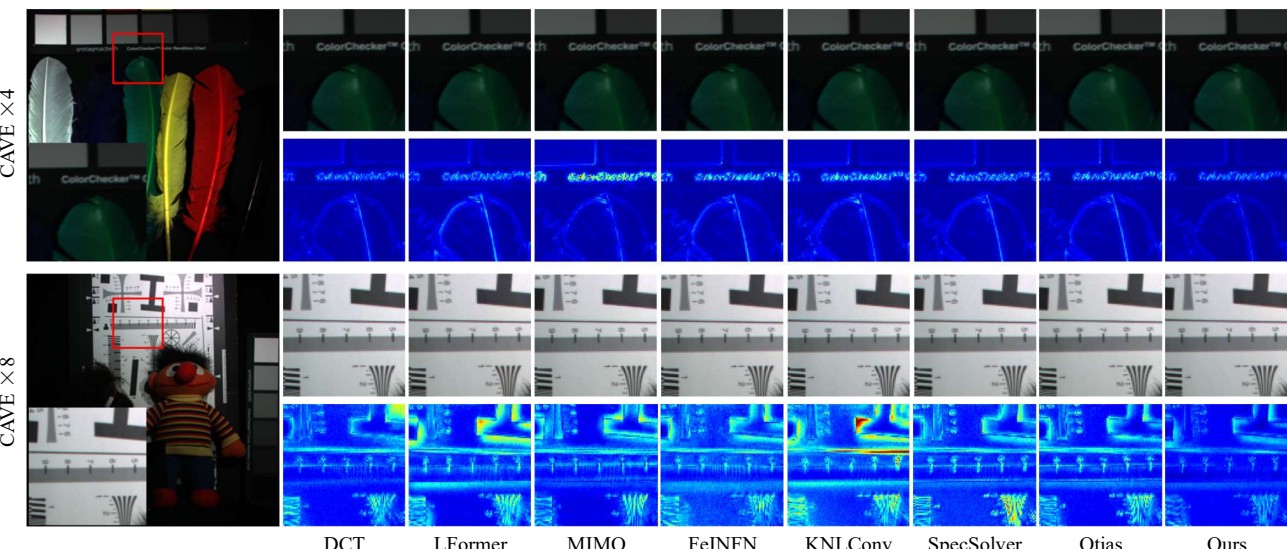

*Figure 3.* Using pseudo-color representation, the upper and lower parts respectively show ×4 and ×8 fusion examples from CAVE. Red rectangles highlight close-up shots. The residuals between GT and the fused results are also given.

the model for 500 epochs. For quantitative evaluation, we report three widely used metrics: Spectral Angle Mapper (SAM) (Yuhas et al., 1992), the Erreur Relative Globale Adimensionnelle de Synthèse (ERGAS) (Wald, 2002), the Peak Signal-to-Noise Ratio (PSNR) (Hore & Ziou, 2010). More detailed settings are given in Appendix C.

### 4.2. Comparison with Leading Methods

We compare our proposal with various state-of-the-art methods, including DCT (Ma et al., 2024), LFormer (Hou et al., 2024), MIMO (Fang et al., 2024), FeINFN (Liang et al., 2024), KNLConv (Ran et al., 2024), SpecSolver (Li et al., 2025d), and Otias (Deng et al., 2025).

**Results on CAVE Dataset.** Specific to the CAVE results reported in Table 1, our method achieves the best performance across all metrics in the ×4 setting. Compared with lead-

ing approaches, it improves PSNR by 0.3%, reduces SAM by 4.4%, and reduces ERGAS by 2.4%. This advantage becomes even more pronounced in the ×8 setting. We attribute this to the fact that many existing methods are tightly coupled to the coordinate domain, making their learned mappings resolution-dependent; as the resolution gap increases, their reconstruction quality degrades more noticeably. In contrast, our method decouples coordinate information from latent features by performing fusion in the spectral latent space and only reconstructing back to the image space afterward, which mitigates performance degradation under large scaling factors. To further demonstrate the strengths of our approach, we provide visual comparisons in Fig. 3, including zoomed-in regions and error maps that highlight fine-grained details. Our fusion results align most closely with the ground truth (GT), producing the highest-quality reconstructions. In the error maps, bluer regions indicate

*Table 2.* The average and standard deviation calculated for all the methods on the Harvard dataset simulating scaling factors of 4 and 8. The best results are in bold, second-best in underline. Params and FLOPs are reported in millions (M) and billions (G), respectively. $*$ indicates $p < 0.01$, and $\dagger$ indicates $p < 0.05$.

| Methods | Params | FLOPs | Harvard $\times 4$ | | | Harvard $\times 8$ | | |
|---|---|---|---|---|---|---|---|---|
| | | | PSNR $\uparrow$ | SAM $\downarrow$ | ERGAS $\downarrow$ | PSNR $\uparrow$ | SAM $\downarrow$ | ERGAS $\downarrow$ |
| Bicubic | - | - | 37.173±3.955 | 2.645±0.700 | 5.445±2.389 | 34.051±3.978 | 3.186±0.883 | 7.587±3.198 |
| DCT$_{2024}$ | 8.15 | 2457.43 | 49.068±2.792* | 2.163±0.543† | 1.756±0.758* | 47.836±3.400* | 2.578±0.678* | 2.088±1.108* |
| LFormer$_{2024}$ | 2.28 | 306.94 | 48.938±2.947* | 2.182±0.567* | 1.706±0.642* | 46.651±3.279* | 2.742±0.741* | 2.305±1.133* |
| MIMO$_{2024}$ | 4.98 | **49.08** | 49.019±2.707* | 2.179±0.544* | 1.688±0.667* | 47.839±3.304* | 2.475±0.691* | 1.958±0.897* |
| FeINFN$_{2024}$ | 3.17 | 382.72 | 49.001±2.944* | 2.155±0.559* | 1.683±0.597* | 47.833±3.433* | 2.466±0.711* | 1.933±0.788* |
| KNLConv$_{2024}$ | **1.73** | 114.04 | 48.536±2.930* | 2.203±0.573* | 1.940±0.948* | 46.706±3.825* | 2.588±0.743* | 2.487±1.441* |
| SpecSolver$_{2025}$ | 3.10 | 364.75 | 48.919±2.827* | 2.169±0.565* | 1.771±0.737* | 47.652±3.306* | 2.533±0.710* | 2.052±0.964* |
| Otias$_{2025}$ | 2.99 | 278.35 | 49.046±2.905* | 2.151±0.561* | 1.728±0.657* | 47.848±3.309† | 2.459±0.680* | 1.907±0.725* |
| Ours | 1.94 | 222.85 | **49.182±2.924** | **2.144±0.550** | **1.658±0.609** | **47.957±3.340** | **2.438±0.671** | **1.879±0.732** |
| Ours vs. Best compe. | | | ▲0.114 (0.2%) | ▼0.019 (0.9%) | ▼0.025 (1.5%) | ▲0.109 (0.2%) | ▼0.021 (0.9%) | ▼0.028 (1.5%) |

*Figure 4.* Using pseudo-color representation, the upper and lower parts respectively show $\times 4$ and $\times 8$ fusion examples from Harvard. Red rectangles highlight close-up shots. The residuals between GT and the fused results are also given.

smaller errors and thus closer agreement with the GT. Compared with other top-performing methods, our error maps exhibit consistently lower errors, especially around edges and textures, indicating superior recovery of high-frequency spatial details. To assess spectral consistency, we further plot the band-wise error of a representative pixel at position (200, 300) in Fig. 5 (left). The error curve of our method stays closest to zero across bands, suggesting improved spectral fidelity.

**Results on Harvard Dataset.** Note that Harvard contains larger-scale scenes than CAVE, implying more complex spatial structures. This setting makes the advantages of our design more apparent, e.g. decoupling latent features from the coordinate domain. As shown in Table 2, LSO consistently outperforms all competing methods across all metrics under both the $\times 4$ and $\times 8$ settings. For instance, in the $\times 4$ case, our PSNR exceeds that of the top three competitors by 0.114 dB, 0.134 dB, and 0.163 dB, respectively. To better illustrate these performance gaps, Fig. 4 presents qualitative

comparisons of the fused images and corresponding error maps. The results further confirm that our method preserves high-fidelity textures and fine details. Moreover, the band-wise error curves in Fig. 5 (right) demonstrate that LSO maintains excellent spectral consistency.

**Statistical Significance Tests.** For each competing method in Tables 1 and 2, we perform 10 independent runs. In each run, we first compute the mean and variance of each metric across all test images, and then report the averages of these run-level statistics over the 10 runs. Because the restoration difficulty differs markedly from image to image, the measured standard deviations can be relatively large. We use $*$ and $\dagger$ to indicate statistical significance at $p < 0.01$ and $p < 0.05$, respectively. Many of the resulting $p$-values fall below 0.01, providing strong statistical evidence. As an illustrative case, consider PSNR on Harvard $\times 4$ when comparing LSO with the second-best method DCT. Although the average PSNR improvement is only 0.114 dB, the run-wise PSNR differences have a standard deviation of 0.045.

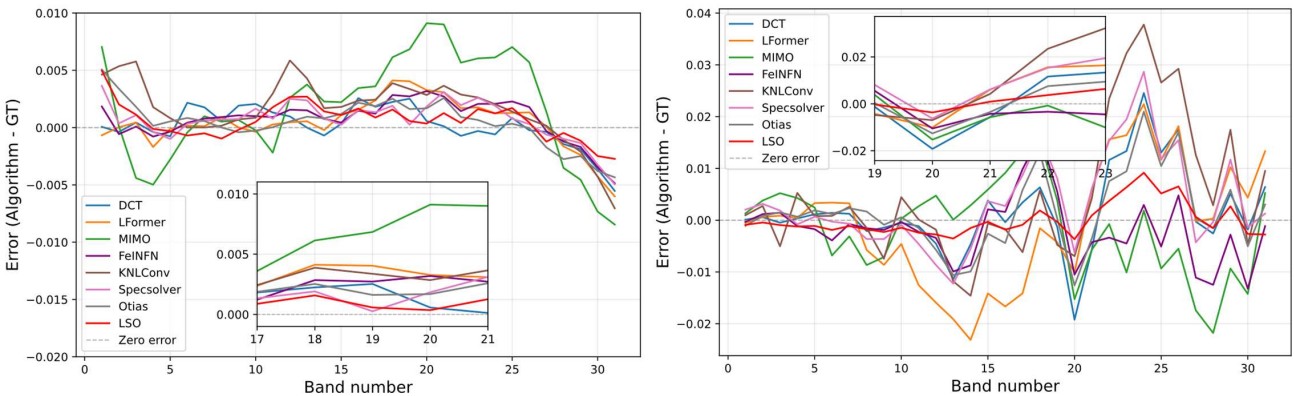

*Figure 5.* Spectral vectors error of the ground truth and various recoveries. Left: CAVE; Right: Harvard.

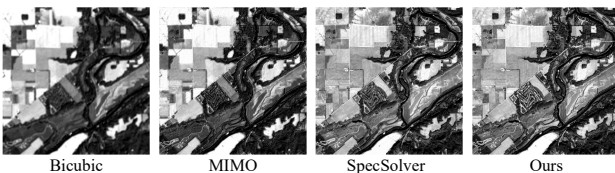

*Figure 6.* Real HSI recovery produced by blind methods.

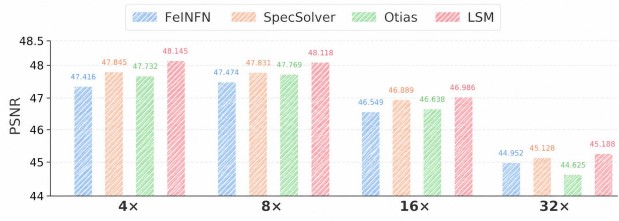

*Figure 7.* PSNR vs Magnification.

A paired, two-sided $t$-test across the 10 runs ($df = 9$) thus yields $p \ll 0.01$, allowing us to reject the null hypothesis of no difference and supporting the conclusion that the gain is real rather than driven by random variation.

**Generalization Analysis.** To investigate LSO's robustness to domain shift, we deploy the model trained exclusively on the CAVE dataset and directly perform SSF inference on real-world hyperspectral images. Since the ground-truth HSI is unavailable, we rely on qualitative visualizations to assess performance. Fig. 6 presents the fusion results of competing blind methods in this setting. Evidently, our method produces the sharpest and most visually distinct reconstructions, whereas other methods appear noticeably hazy and lack clear structural details.

Furthermore, we compare the performance of different algorithms trained on the $8\times$ CAVE dataset and tested at $4\times$, $16\times$, and $32\times$ scaling factors, as shown in the Fig. 7. FeINFN and OTIAS employ INR to model the underlying continuous spectral functions, thereby exhibiting strong generalization across different scaling factors. Moreover, by incorporating semantic information, SpecSolver demonstrates even better generalization than these INR-based methods.

*Table 3.* Performance across different scales.

| Scale | PSNR ↑ | SAM ↓ | ERGAS ↓ | Params ↓ | FLOPs ↓ |
|---|---|---|---|---|---|
| 1 | 42.462 | 4.268 | 3.756 | 1.534M | 200.478G |
| 2 | 47.859 | 3.201 | 2.379 | 1.739M | 218.376G |
| 3 | 49.182 | 2.144 | 1.658 | 1.943M | 222.850G |
| 4 | 49.211 | 2.141 | 1.633 | 2.147M | 223.969G |

Different from both lines of work, our approach takes an alternative route: by leveraging the latent space to explicitly decouple coordinate information from features, it substantially mitigates performance degradation under resolution changes.

**Ablation Study.** As discussed in Section 3.2, the hierarchical patch-based design is introduced to provide indispensable multi-scale information for the SSF task. To verify its effectiveness, we conduct the following ablation studies. As shown in Table 3, $scale$ denotes the number of scales. When $scale = 1$, using a single scale substantially degrades the reconstruction performance. As the number of scales increases, the performance improves progressively, at the cost of higher parameter count and computational complexity. Considering this trade-off, we set $scale = 3$ as the default hyperparameter throughout this paper. Notably, the FLOPs increase becomes marginal beyond $scale = 3$, since the additional coarser scales operate on much smaller feature maps and thus contribute little extra computation. Please refer to Appendix D for additional ablation and other results.

## 5. Conclusion

In this work, we presented **Latent Spectral Operators (LSO)**, a latent-space framework for spatial-spectral fusion. LSO first employs a *cross-attention projection* in which learned latent tokens act as spectral prompts to compress high-dimensional observations into a compact latent representation, and then integrates rich multi-scale cues via a hierarchical, patch-based architecture. To parameterize the latent fusion mapping in a structured and controllable manner, we further introduced a *Trigonometric Basis Solver*

that represents the latent operator using a basis expansion, naturally enabling multi-frequency modeling with an explicit capacity-stability trade-off governed by the number of basis functions. Extensive experiments on the benchmarks demonstrate that LSO achieves consistent state-of-the-art accuracy with statistically significant improvements, while remaining computationally efficient and exhibiting strong cross-scale transferability. Regarding limitations, our current evaluation still has two boundaries. First, as stated in Sec. 4.1, supervised experiments on CAVE and Harvard follow the standard simulation protocol because real paired HR-HSI ground truth is unavailable; therefore, the quantitative benchmarks are based on synthetically degraded observations rather than fully real acquisition pairs. Second, for the real-world setting, the absence of ground-truth HSI means that evaluation is mainly qualitative, which limits the strength of conclusions about absolute reconstruction fidelity.

## Acknowledgements

This work was supported in part by the National Natural Science Foundation of China under Grant 62276232 and the Key Program of Natural Science Foundation of Zhejiang Province under Grant LZ24F030012.

## Impact Statement

We will clarify that hyperspectral reconstruction can support beneficial applications such as change detection, disaster assessment, and geolocation, but these same capabilities may also be misused in surveillance-oriented or privacy-sensitive remote-sensing scenarios.

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

# Appendix

## A. Model Architecture

*Table 4.* Symbol table for the downsampling, upsampling, and patchification operations.

| Symbol | Name | Meaning / Description |
|---|---|---|
| $k$ | Scale index | Index of the multi-scale hierarchy, $k = 1, \ldots, K$. |
| $K$ | Scales | Total number of scales. |
| $\Omega^k$ | Coordinate set (domain) | The coordinate set (spatial domain) at the $k$-th scale; $x \in \Omega^k$. |
| $x, x'$ | Coordinate | A spatial coordinate/location in the domain $\Omega^k$. |
| $\{\cdot\}_{x \in \Omega^k}$ | Feature field | A set (field) of features defined over all coordinates $x \in \Omega^k$. |
| $\boldsymbol{Z^k}(x)$ | Deep representation | Deep feature representation at coordinate $x$ on the $k$-th scale. |
| $\boldsymbol{Z^{k+1}}(x)$ | Downsampled representation | Feature representation after downsampling from scale $k$ to $k+1$, defined on $\Omega^{k+1}$. |
| $\boldsymbol{Y^k}(x)$ | Coordinate-space feature | Coordinate-space feature at coordinate $x$ on the $k$-th scale (typically used in decoding/upsampling). |
| $\boldsymbol{Y^{k+1}}(x)$ | Coarser feature | Coordinate-space feature at coordinate $x$ on the $(k+1)$-th scale. |
| $\mathrm{MaxPool}(\cdot)$ | Max pooling | Aggregates local observations by maximum pooling (downsampling in spatial resolution). |
| $\mathrm{Conv}(\cdot)$ | Convolution | Local convolution for feature transformation/fusion. |
| $\mathrm{Interpolation}(\cdot)$ | Interpolation | Upsampling by interpolation (bilinear interpolation is adopted in the paper). |
| $\mathrm{Concat}([\cdot, \cdot])$ | Concatenation | Concatenates two features (typically along the channel dimension) before fusion. |
| $[\cdot, \cdot]$ | Input list | A list/tuple notation indicating two feature inputs to be concatenated. |
| $\{\Omega_j^k\}_{j=1}^{P_k}$ | Patch set | A set of non-overlapping patches (local coordinate subsets) at the $k$-th scale. |
| $\Omega_j^k$ | Patch (region) | The $j$-th patch (local coordinate subset) at the $k$-th scale; each patch contains an equal number of coordinates. |
| $P_k$ | Patches | Number of patches at the $k$-th scale. |
| $\mathrm{Patchify}(\Omega^k)$ | Patchify | Splits $\Omega^k$ into $\{\Omega_j^k\}_{j=1}^{P_k}$ (non-overlapping local regions). |
| $\mathrm{De\text{-}Patchify}(\{\Omega_j^k\}_{j=1}^{P_k})$ | De-Patchify | Splices the patch set back to the full coordinate set $\Omega^k$. |

In this section, we illustrate the downsampling and patchification operations to facilitate a clearer understanding of the mathematical notation in Table 4.

**Downsample.** Given deep representations $\{\boldsymbol{Z^k}(x)\}_{x \in \Omega^k}$ at the $k$-th scale, the downsample operation aggregates local observations with maximum pooling and convolution, which can be formulized as:

$$\{\boldsymbol{Z^{k+1}}(x)\}_{x \in \Omega^{k+1}} = \mathrm{Conv}\Big(\mathrm{MaxPool}\big(\{\boldsymbol{Z^k}(x)\}_{x \in \Omega^k}\big)\Big), \quad k = 1, \ldots, (K-1). \tag{8}$$

**Upsample.** Given the coordinate-space features $\{\boldsymbol{Y^{k+1}}(x)\}_{x \in \Omega^{k+1}}$ and $\{\boldsymbol{Y^k}(x)\}_{x \in \Omega^k}$ at the $(k+1)$-th and $k$-th scales respectively, the upsample process fuses the interpolated $(k+1)$-th features and the $k$-th features with local convolution:

$$\{\boldsymbol{Y^k}(x)\}_{x \in \Omega^k} = \mathrm{Conv}\Big(\mathrm{Concat}\Big(\big[\mathrm{Interpolation}(\{\boldsymbol{Y^{k+1}}(x)\}_{x \in \Omega^{k+1}}), \{\boldsymbol{Y^k}(x)\}_{x \in \Omega^k}\big]\Big)\Big), \quad k = (K-1), \ldots, 1. \tag{9}$$

where we adopt the bilinear $\mathrm{Interpolation}(\cdot)$ in the paper.

**Patchify and De-Patchify.** The patchify operation splits the coordinate set into several non-overlapping local regions with an equal number of coordinates. At the $k$-th scale, this process is formulized as:

$$\{\Omega_j^k\}_{j=1}^{P_k} = \mathrm{Patchify}(\Omega^k). \tag{10}$$

The de-patchify operation splices the patches back, i.e.,

$$\Omega^k = \mathrm{De\text{-}Patchify}\big(\{\Omega_j^k\}_{j=1}^{P_k}\big). \tag{11}$$

# B. Theoretical basis

Please refer to the notation in Table 5 for a clearer understanding of the related theory.

*Table 5.* Symbol table for Appendix B.1 (Proof of Remark 3.1).

| Symbol | Meaning |
|---|---|
| *Proof of Remark 3.1: discretized coordinate space and Fourier approximation* | |
| $\hat{\Omega}$ | Underlying continuous spatial domain (conceptual). |
| $\Omega = \{x_1, \ldots, x_u\} \subset \hat{\Omega}$ | Finite sampling grid (pixel coordinates). |
| $x_j$ | The $j$-th sampled coordinate in $\Omega$. |
| $u := |\Omega|$ | Number of sampled coordinates; $u = H \times W$ (or $u = H_k \times W_k$ at scale $k$). |
| $H, W$ | Spatial height/width at the finest scale. |
| $H_k, W_k$ | Spatial height/width at scale $k$ in a multiscale hierarchy. |
| $C$ | Channel dimension of the spectral field / feature. |
| $z \in \mathcal{Z}(\hat{\Omega}; \mathbb{R}^C)$ | Continuous spectral field (vector-valued function over $\hat{\Omega}$). |
| $\boldsymbol{Z} = [z(x_1), \ldots, z(x_u)]^\top \in \mathbb{R}^{u \times C}$ | Discrete samples of $z$ on $\Omega$. |
| $\mathcal{F} : \mathcal{Z}(\hat{\Omega}; \mathbb{R}^C) \to \mathcal{Y}(\hat{\Omega}; \mathbb{R}^C)$ | Continuous operator (ideal SSF mapping). |
| $\tilde{g} : \mathbb{R}^{u \times C} \to \mathbb{R}^{u \times C}$ | Discretized induced map on sampled tensors. |
| $\ell$ | Output-channel index for channel slicing. |
| $\boldsymbol{Z}_{-\ell} \in \mathbb{R}^{u \times (C-1)}$ | Reference values for all channels except the $\ell$-th one. |
| $\tilde{g}_\ell(\cdot) \in \mathbb{R}^u$ | The $\ell$-th output channel of $\tilde{g}$, vectorized over $\Omega$. |
| $g_\ell : \mathbb{R}^u \to \mathbb{R}^u$ | Slice map varying only the $\ell$-th input channel: $g_\ell(z) = \tilde{g}_\ell([z, \boldsymbol{Z}_{-\ell}])$. |
| $[-\pi, \pi)^u$ | Bounded/normalized input domain for analysis. |
| $\bar{g}_\ell$ | $2\pi$-periodic extension of $g_\ell$ onto $\mathbb{T}^u$. |
| $\mathbb{T}^u := (\mathbb{R}/2\pi\mathbb{Z})^u$ | $u$-dimensional torus (periodic domain). |
| $p$ | Exponent in $L^p$ spaces; $1 \le p \le \infty$, $p \ne 2$. |
| $L^p(\mathbb{T}^u; \mathbb{R}^u)$ | $\mathbb{R}^u$-valued (Bochner) $L^p$ functions on $\mathbb{T}^u$. |
| $A$ | Truncation order of the $u$-variate Fourier partial sum. |
| $k \in \mathbb{Z}^u$ | Multi-index of Fourier frequencies. |
| $\|k\|_\infty \le A$ | Cubic truncation set of multi-frequencies. |
| $\widehat{\bar{g}}_\ell(k)$ | Fourier coefficient of $\bar{g}_\ell$ (defined component-wise by integration). |
| $\bar{g}_\ell^{(A)}$ | $u$-variate Fourier partial sum of order $A$. |
| $\| \cdot \|_p$ | $\ell^p$ norm on vectors in $\mathbb{R}^u$. |
| $\| \cdot \|_{L^p(\mathbb{T}^u)}$ | $L^p$ norm over the torus $\mathbb{T}^u$. |
| $K_1, K_2$ | Lipschitz and approximation constants (independent of $A$). |

## B.1. Proof of Remark 3.1

**Assumption B.1** (Finite sampling grid). In practical spatial–spectral fusion, both the low-resolution hyperspectral image and the high-resolution multispectral image are observed only on a finite pixel grid. For theoretical derivations we assume the underlying continuous domain $\hat{\Omega}$ is sampled by a finite set

$$\Omega = \{x_1, \ldots, x_u\} \subset \hat{\Omega}, \qquad u := |\Omega| < \infty, \tag{12}$$

where $u = H \times W$ for an $H \times W$ image (or $u = H_k \times W_k$ at scale $k$ in the multiscale setting). Accordingly, any spectral field $z \in \mathcal{Z}(\hat{\Omega}; \mathbb{R}^C)$ is represented by its samples $\boldsymbol{Z} = [z(x_1), \ldots, z(x_u)]^\top \in \mathbb{R}^{u \times C}$, and the continuous operator $\mathcal{F} : \mathcal{Z}(\hat{\Omega}; \mathbb{R}^C) \to \mathcal{Y}(\hat{\Omega}; \mathbb{R}^C)$ induces a discretized map $\tilde{g} : \mathbb{R}^{u \times C} \to \mathbb{R}^{u \times C}$.

**Assumption B.2** (Normalization and periodic extension). Fix an output channel $\ell$ and consider a restricted (slice) map $g_\ell : \mathbb{R}^u \to \mathbb{R}^u$ defined from $\tilde{g}$ by varying only the $\ell$-th input channel while keeping the remaining channels fixed at a reference value $\boldsymbol{Z}_{-\ell} \in \mathbb{R}^{u \times (C-1)}$:

$$g_\ell(z) := \tilde{g}_\ell\big([z, \boldsymbol{Z}_{-\ell}]\big), \tag{13}$$

where $\tilde{g}_\ell(\cdot) \in \mathbb{R}^u$ denotes the $\ell$-th output channel vectorized over $\Omega$. Since discrete samples are bounded in practice (after standard normalization), we restrict $z \in [-\pi, \pi)^u$. To leverage tools from trigonometric approximation, we consider the

$2\pi$-periodic extension $\bar{g}_\ell$ of $g_\ell$ from $[-\pi, \pi)^u$ to the torus $\mathbb{T}^u := (\mathbb{R}/2\pi\mathbb{Z})^u$. This periodic extension is only a technical device for analysis and does not assert that the true SSF mapping is periodic.

**Theorem B.3** (Dyachenko-inspired convergence in the discretized coordinate space (Dyachenko, 1995)). *Let $\Omega = \{x_j\}_{j=1}^u$ be a finite sampling grid ($u \geq 2$). Under Assumptions B.1–B.2, suppose $\bar{g}_\ell \in L^p(\mathbb{T}^u; \mathbb{R}^u)$ with $1 \leq p \leq \infty$, $p \neq 2$. Define the $u$-variate Fourier partial sum of order $A$:*

$$\bar{g}_\ell^{(A)}(z) = \sum_{\substack{k \in \mathbb{Z}^u \\ \|k\|_\infty \leq A}} \widehat{\bar{g}_\ell}(k) \, e^{ik^\top z}, \qquad \widehat{\bar{g}_\ell}(k) := \frac{1}{(2\pi)^u} \int_{[-\pi,\pi)^u} \bar{g}_\ell(t) \, e^{-ik^\top t} \, dt, \tag{14}$$

*where the integral is taken component-wise.*

*If $\bar{g}_\ell$ is Lipschitz in $\ell^p$, i.e., there exists $K_1 \geq 0$ such that*

$$\|\bar{g}_\ell(z) - \bar{g}_\ell(z')\|_p \leq K_1 \|z - z'\|_p, \qquad \forall z, z' \in [-\pi, \pi)^u, \tag{15}$$

*and if $(u-1)\left|\frac{1}{2} - \frac{1}{p}\right| < 1$, then there exists $K_2 > 0$ independent of $A$ such that*

$$\|\bar{g}_\ell - \bar{g}_\ell^{(A)}\|_{L^p(\mathbb{T}^u)} \leq K_2 \, A^{(u-1)\left|\frac{1}{2} - \frac{1}{p}\right| - 1}. \tag{16}$$

*Remark* B.4 (Interpretation and scope). The condition $(u-1)\left|\frac{1}{2} - \frac{1}{p}\right| < 1$ is a technical requirement in Dyachenko's result that ensures the stated $L^p$ convergence rate. For high-resolution images, $u = H \times W$ can be very large, and the condition may fail unless $p$ is extremely close to 2. Our use of Theorem B.3 is therefore *illustrative*: it shows that even in a favorable regime where such a rate is guaranteed, the exponent explicitly deteriorates with $u$, highlighting the curse of dimensionality in the discretized coordinate space.

**Corollary B.5** (Dimension dependence). *Under the assumptions of Theorem B.3, the approximation rate $\|\bar{g}_\ell - \bar{g}_\ell^{(A)}\|_{L^p(\mathbb{T}^u)} = \mathcal{O}\left(A^{(u-1)\left|\frac{1}{2} - \frac{1}{p}\right| - 1}\right)$ deteriorates as $u$ increases.*

This corresponds to Remark 3.1 in the main text. Next, we introduce the theory related to Remark 3.2.

## B.2. Proof of Remark 3.2

**Assumption B.6** (Bounded token channels and normalization). For each channel coordinate $r \in \{1, \ldots, d_{\text{latent}}\}$, assume the token entries are (channel-wise) bounded: there exist constants $m_r < M_r$ such that $(T_Z)_{i,r} \in [m_r, M_r]$ for all token indices $i$. Define an invertible affine normalization $\mathcal{N}_r : [m_r, M_r] \to [0, \pi]$ by

$$\mathcal{N}_r(t) = \pi \cdot \frac{t - m_r}{M_r - m_r}, \qquad \mathcal{N}_r^{-1}(x) = m_r + \frac{M_r - m_r}{\pi} x. \tag{17}$$

Denote the normalized scalar input by $x_{i,r} := \mathcal{N}_r\big((T_Z)_{i,r}\big) \in [0, \pi]$. Since $\mathcal{N}_r^{-1}$ is affine, we use the same expression as its natural extension to $\mathbb{R}$ (and it is $(M_r - m_r)/\pi$-Lipschitz on $\mathbb{R}$).

**Lemma B.7** (Token-wise reduction to 1D mappings). *Assume* Solve *is implemented by a finite trigonometric expansion applied* token-wise *on normalized inputs, with a residual connection. Namely, for each channel coordinate $r$ there exist coefficients $\{b_r, a_{r,q}, c_{r,q}\}_{q=1}^Q$ such that for all tokens $i$,*

$$(T_Y)_{i,r} = \mathcal{N}_r^{-1}\left(f_r^{(Q)}(x_{i,r})\right), \qquad f_r^{(Q)}(x) = x + h_r^{(Q)}(x), \tag{18}$$

*where $x_{i,r} \in [0, \pi]$ and*

$$h_r^{(Q)}(x) = b_r + \sum_{q=1}^Q a_{r,q} \sin(qx) + \sum_{q=1}^Q c_{r,q} \cos(qx). \tag{19}$$

*Then, for each fixed $r$, the action of* Solve *on the $r$-th channel coordinate reduces to a scalar mapping $f_r^{(Q)} : [0, \pi] \to \mathbb{R}$ applied entry-wise across tokens. Moreover, the residual part $f_r^{(Q)}(x) - x = h_r^{(Q)}(x)$ is a degree-$Q$ trigonometric polynomial in $x$.*

*Table 6.* Symbol table for Appendix B.2 (Proof of Remark 3.2).

| Symbol | Name | Meaning / Description |
|---|---|---|
| $i$ | Token index | Index of tokens (rows) in the token matrix. |
| $r$ | Channel coordinate | Channel (feature dimension) index, $r \in \{1, \ldots, d_{\text{latent}}\}$. |
| $d_{\text{latent}}$ | Latent dimension | Number of channels (latent feature dimensions) of the token representation. |
| $\boldsymbol{T_Z}$ | Input token matrix | Input token matrix to Solve (before mapping), where $(\boldsymbol{T_Z})_{i,r}$ is the entry of token $i$ in channel $r$. |
| $\boldsymbol{T_Y}$ | Output token matrix | Output token matrix after applying Solve, where $(\boldsymbol{T_Y})_{i,r}$ is the mapped entry of token $i$ in channel $r$. |
| $\boldsymbol{T_Y^\star}$ | Target token matrix | Target (ground-truth) token matrix, where $(\boldsymbol{T_Y^\star})_{i,r}$ is the desired entry of token $i$ in channel $r$. |
| $(\boldsymbol{T_Z})_{i,r}$ | Token entry | Scalar entry of $\boldsymbol{T_Z}$ at token $i$ and channel $r$. |
| $(\boldsymbol{T_Y})_{i,r}$ | Token entry | Scalar entry of $\boldsymbol{T_Y}$ at token $i$ and channel $r$. |
| $(\boldsymbol{T_Y^\star})_{i,r}$ | Target entry | Scalar entry of $\boldsymbol{T_Y^\star}$ at token $i$ and channel $r$. |
| $m_r, M_r$ | Channel bounds | Lower/upper bounds for the $r$-th channel entries, satisfying $(\boldsymbol{T_Z})_{i,r} \in [m_r, M_r]$. |
| $\mathcal{N}_r(\cdot)$ | Normalization | Invertible affine normalization $\mathcal{N}_r : [m_r, M_r] \to [0, \pi]$. |
| $\mathcal{N}_r^{-1}(\cdot)$ | Inverse normalization | Inverse affine mapping $\mathcal{N}_r^{-1} : [0, \pi] \to [m_r, M_r]$, naturally extended to $\mathbb{R}$; Lipschitz constant $(M_r - m_r)/\pi$. |
| $x_{i,r}$ | Normalized scalar input | Normalized token entry $x_{i,r} := \mathcal{N}_r\big((\boldsymbol{T_Z})_{i,r}\big) \in [0, \pi]$. |
| Solve | Token-wise solver | A token-wise mapping implemented by a finite trigonometric expansion with a residual connection. |
| $f_r(\cdot)$ | Ground-truth mapping | Unknown channel-wise target function on $[0, \pi]$ defined by $\mathcal{N}_r\big((\boldsymbol{T_Y^\star})_{i,r}\big) = f_r(x_{i,r})$. |
| $f_r^{(Q)}(\cdot)$ | Trigonometric approximant | Degree-$Q$ approximating scalar mapping parameterized by the model, of the form $f_r^{(Q)}(x) = x + h_r^{(Q)}(x)$. |
| $Q$ | Truncation degree | Maximum frequency / degree of the trigonometric expansion (number of harmonics). |
| $h_r^{(Q)}(\cdot)$ | Residual term | Degree-$Q$ trigonometric polynomial representing the residual $f_r^{(Q)}(x) - x$. |
| $b_r$ | Bias coefficient | Constant term in the trigonometric expansion for channel $r$. |
| $a_{r,q}$ | Sine coefficient | Coefficient of $\sin(qx)$ for channel $r$ and harmonic $q$. |
| $c_{r,q}$ | Cosine coefficient | Coefficient of $\cos(qx)$ for channel $r$ and harmonic $q$. |
| $q$ | Harmonic index | Frequency index in the expansion, $q = 1, \ldots, Q$. |
| $g_r(\cdot)$ | Residual function | Residual w.r.t. identity: $g_r(x) := f_r(x) - x$ on $[0, \pi]$. |
| $\tilde{g}_r(\cdot)$ | Even extension | Even extension of $g_r$ to $[-\pi, \pi]$ defined by $\tilde{g}_r(x) := g_r(|x|)$. |
| $\bar{g}_r(\cdot)$ | Periodic extension | $2\pi$-periodic extension of $\tilde{g}_r$ to $\mathbb{R}$ (a function on $\mathbb{T} = \mathbb{R}/2\pi\mathbb{Z}$). |
| $\mathbb{T}$ | 1D torus | The periodic domain $\mathbb{R}/2\pi\mathbb{Z}$ for $2\pi$-periodic functions. |
| $d_{\mathbb{T}}(\cdot, \cdot)$ | Geodesic distance | Geodesic distance on $\mathbb{T}$ used to define Lipschitz continuity and modulus of continuity. |
| $\phi(\cdot)$ | Generic periodic function | A generic continuous $2\pi$-periodic function on $\mathbb{T}$ used in the Jackson-type inequality. |
| $\omega(\phi, \delta)$ | Modulus of continuity | $\omega(\phi, \delta) := \sup_{d_{\mathbb{T}}(x,y) \leq \delta} |\phi(x) - \phi(y)|$. |
| $\deg(p)$ | Trigonometric degree | Degree (maximum frequency) of a trigonometric polynomial $p$. |
| $p_{r,Q}(\cdot)$ | Trig polynomial | Degree-$Q$ trigonometric polynomial approximating $\bar{g}_r$ in the Jackson bound. |
| $\alpha_{r,0}$ | Constant coefficient | Constant term in $p_{r,Q}(x)$. |
| $\alpha_{r,q}$ | Sine coefficient | Coefficient of $\sin(qx)$ in $p_{r,Q}(x)$. |
| $\beta_{r,q}$ | Cosine coefficient | Coefficient of $\cos(qx)$ in $p_{r,Q}(x)$. |
| $L_r$ | Lipschitz constant | Lipschitz constant of $f_r$ on $[0, \pi]$: $|f_r(x) - f_r(y)| \leq L_r|x - y|$. |
| $\text{Lip}(\cdot)$ | Lipschitz seminorm | Lipschitz constant/operator of a function on the corresponding domain. |
| $C$ | Jackson constant | Universal constant in the Jackson-type inequality $\inf_{\deg(p) \leq Q} \sup_{x \in \mathbb{T}} |\phi(x) - p(x)| \leq C\,\omega(\phi, 1/Q)$. |
| $K$ | Universal constant | Universal constant in the final approximation bound (after absorbing universal factors). |
| inf | Infimum | Best (optimal) approximation error over a function class. |
| sup | Supremum | Uniform (worst-case) error over the domain. |
| $\pi$ | Pi | Mathematical constant defining the normalization range $[0, \pi]$. |

*Proof.* Equation (18) shows that for each fixed channel coordinate $r$, the output entry $(\boldsymbol{T_Y})_{i,r}$ depends on the input only through the scalar $x_{i,r}$ via the same 1D mapping $f_r^{(Q)}$, followed by the affine inverse normalization $\mathcal{N}_r^{-1}$; hence the mapping is applied entry-wise across tokens. Moreover, $h_r^{(Q)}$ is a finite linear combination of $\{\sin(qx), \cos(qx)\}_{q=1}^{Q}$ plus a constant term, and therefore is a degree-$Q$ trigonometric polynomial.

To relate this parameterization to a target mapping, we may define an (unknown) channel-wise function $f_r : [0, \pi] \to \mathbb{R}$ by $\mathcal{N}_r((\boldsymbol{T_Y^\star})_{i,r}) = f_r(x_{i,r})$ for all tokens $i$. Then learning Solve amounts to selecting the coefficients in $f_r^{(Q)}$ to approximate $f_r$. $\qquad\square$

**Theorem B.8** (Jackson-type 1D approximation guarantee for Lipschitz channel mappings). *Let $f_r : [0, \pi] \to \mathbb{R}$ be Lipschitz continuous with constant $L_r$, i.e., $|f_r(x) - f_r(y)| \le L_r|x - y|$ for all $x, y \in [0, \pi]$. Then for any $Q \ge 1$, there exist coefficients $\{b_r, a_{r,q}, c_{r,q}\}_{q=1}^{Q}$ such that the model $f_r^{(Q)}(x) = x + h_r^{(Q)}(x)$ in (18), with $h_r^{(Q)}$ a degree-$Q$ trigonometric polynomial, satisfies the uniform error bound*

$$\sup_{x \in [0,\pi]} \left| f_r(x) - f_r^{(Q)}(x) \right| \le K \frac{L_r + 1}{Q}, \qquad Q \ge 1, \tag{20}$$

*where $K > 0$ is a universal constant independent of $f_r$ and $Q$.*

*Proof.* **Step 1 (Residual form).** Define the residual function

$$g_r(x) := f_r(x) - x, \qquad x \in [0, \pi].$$

Since $f_r$ is Lipschitz with constant $L_r$, for all $x, y \in [0, \pi]$,

$$|g_r(x) - g_r(y)| = |f_r(x) - f_r(y) - (x - y)| \le |f_r(x) - f_r(y)| + |x - y| \le (L_r + 1)|x - y|.$$

Hence $g_r$ is Lipschitz with $\mathrm{Lip}(g_r) \le L_r + 1$.

**Step 2 (Even extension to $[-\pi, \pi]$).** Define $\tilde{g}_r : [-\pi, \pi] \to \mathbb{R}$ by the even extension

$$\tilde{g}_r(x) := g_r(|x|).$$

Then for any $x, y \in [-\pi, \pi]$,

$$|\tilde{g}_r(x) - \tilde{g}_r(y)| = |g_r(|x|) - g_r(|y|)| \le \mathrm{Lip}(g_r)\,||x| - |y|| \le \mathrm{Lip}(g_r)\,|x - y|.$$

Therefore $\tilde{g}_r$ is Lipschitz on $[-\pi, \pi]$ with $\mathrm{Lip}(\tilde{g}_r) \le \mathrm{Lip}(g_r) \le L_r + 1$.

**Step 3 ($2\pi$-periodic extension).** Extend $\tilde{g}_r$ to a $2\pi$-periodic function $\bar{g}_r$ on $\mathbb{R}$ by periodic repetition, equivalently viewing $\bar{g}_r$ as a function on the 1D torus $\mathbb{T} = \mathbb{R}/2\pi\mathbb{Z}$. Then $\bar{g}_r$ is Lipschitz on $\mathbb{T}$ (with respect to the geodesic distance) and

$$\mathrm{Lip}(\bar{g}_r) \le \mathrm{Lip}(\tilde{g}_r) \le L_r + 1.$$

**Step 4 (Jackson-type trigonometric approximation).** For a continuous $2\pi$-periodic function $\phi$ on $\mathbb{T}$, define its modulus of continuity

$$\omega(\phi, \delta) := \sup_{d_{\mathbb{T}}(x,y) \le \delta} |\phi(x) - \phi(y)|,$$

where $d_{\mathbb{T}}$ is the geodesic distance on $\mathbb{T}$. A classical Jackson-type inequality for trigonometric approximation states that there exists a universal constant $C > 0$ such that for any $Q \ge 1$,

$$\inf_{\deg(p) \le Q} \sup_{x \in \mathbb{T}} |\phi(x) - p(x)| \le C\,\omega(\phi, 1/Q), \tag{21}$$

where the infimum is over all degree-$Q$ trigonometric polynomials $p$. Applying (21) to $\phi = \bar{g}_r$ and using Lipschitz continuity gives

$$\omega(\bar{g}_r, 1/Q) \le \mathrm{Lip}(\bar{g}_r) \cdot \frac{1}{Q} \le \frac{L_r + 1}{Q}.$$

Hence there exists a degree-$Q$ trigonometric polynomial

$$p_{r,Q}(x) = \alpha_{r,0} + \sum_{q=1}^{Q} \alpha_{r,q} \sin(qx) + \sum_{q=1}^{Q} \beta_{r,q} \cos(qx)$$

such that

$$\sup_{x \in [-\pi, \pi]} |\bar{g}_r(x) - p_{r,Q}(x)| \le C \frac{L_r + 1}{Q}.$$

Restricting back to $[0, \pi]$ yields

$$\sup_{x \in [0, \pi]} |g_r(x) - p_{r,Q}(x)| \le C \frac{L_r + 1}{Q}.$$

**Step 5 (Add back the identity / residual).** Define

$$f_r^{(Q)}(x) := x + p_{r,Q}(x), \qquad x \in [0, \pi].$$

Then $f_r^{(Q)}$ has exactly the form in (18) with $b_r = \alpha_{r,0}$, $a_{r,q} = \alpha_{r,q}$, and $c_{r,q} = \beta_{r,q}$. Moreover, for all $x \in [0, \pi]$,

$$|f_r(x) - f_r^{(Q)}(x)| = |g_r(x) - p_{r,Q}(x)| \le C \frac{L_r + 1}{Q}.$$

Renaming $C$ to $K$ proves (20). $\qquad\square$

**Corollary B.9** (Entry-wise approximation for the token matrix). *Under Assumption B.6 and Theorem B.8, for any token index $i$ and channel coordinate $r$, let $(\boldsymbol{T}_Y^\star)_{i,r} := \mathcal{N}_r^{-1}\big(f_r(x_{i,r})\big)$. Then*

$$\begin{aligned}
\big|(\boldsymbol{T}_Y)_{i,r} - (\boldsymbol{T}_Y^\star)_{i,r}\big| &= \Big|\mathcal{N}_r^{-1}\big(f_r^{(Q)}(x_{i,r})\big) - \mathcal{N}_r^{-1}\big(f_r(x_{i,r})\big)\Big| \\
&\le \frac{M_r - m_r}{\pi} \sup_{x \in [0, \pi]} \big|f_r(x) - f_r^{(Q)}(x)\big| \\
&\le \frac{M_r - m_r}{\pi} \cdot K \frac{L_r + 1}{Q}, \qquad Q \ge 1.
\end{aligned} \tag{22}$$

*In other words, the approximation error in the original (unnormalized) token space inherits the same $\mathcal{O}(1/Q)$ rate up to the scaling factor $(M_r - m_r)/\pi$.*

## C. Experiment Details

### C.1. Datasets

*Table 7.* Ground-truth (GT) data shapes for datasets.

| Dataset | Train GT shape | Val GT shape | Test GT shape |
|---------|---------------|--------------|---------------|
| CAVE | (3136, 31, 64, 64) | (784, 31, 64, 64) | (11, 31, 512, 512) |
| Harvard | (3136, 31, 64, 64) | (784, 31, 64, 64) | (10, 31, 1000, 1000) |

As shown in Figs. 8 and 9, we present the pseudo-color representations of the CAVE and Harvard test sets, respectively. And, we summarize the dataset splits in Table 7 for reference. The dataset used in our experiments is publicly available on ModelScope at https://www.modelscope.cn/datasets/WeiLi419/Latent_Spectral_Operators_Cave_and_Harvard to facilitate reproducibility.

For realistic SSF, we use the Sentinel-2 image as HrMSI and image acquired by Hyperion sensor onboard of Earth Observing-1 satellite as LrHSI. The Sentinel-2 MSI has 13 spectral bands and we use the three spectral bands (490, 560, and 665 nm) with a spatial resolution of 10m for SSF. The Hyperion HSI has 220 spectral bands covering the wavelength regions of 400–2500 nm with a spatial resolution of 30m. With the noisy bands discarded, we retain the first 31 spectra for the practical SSF. In our experiments, an area of size $180 \times 180$ from the Hyperion HSI and a corresponding image with the size of $540 \times 540$ from the Sentinel-2 MSI are selected. As shown in Figs. 10 and 11, we provide band-wise grayscale visualizations for each spectral band.

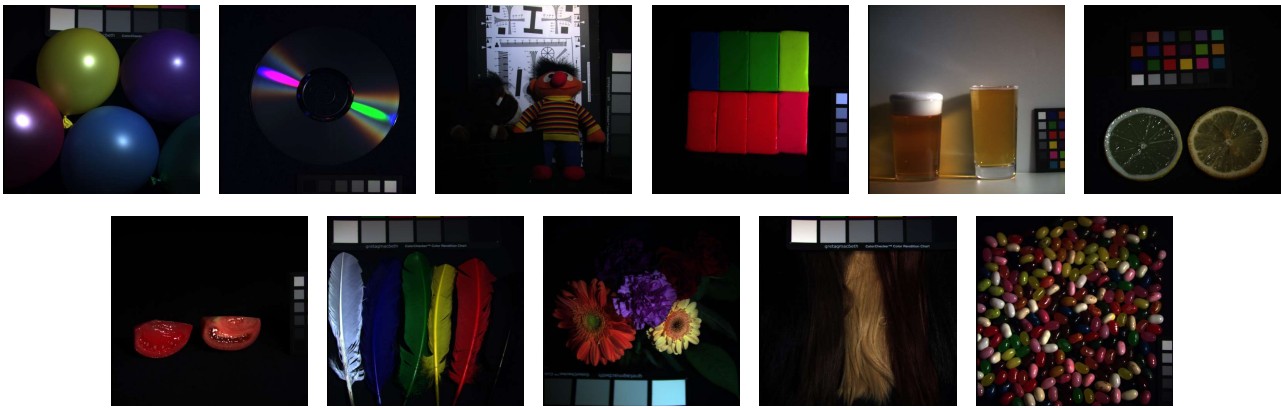

*Figure 8.* The 11 testing images from the CAVE dataset. A pseudo-color representation is used combining the 25th, the 15th, and the 5th bands.

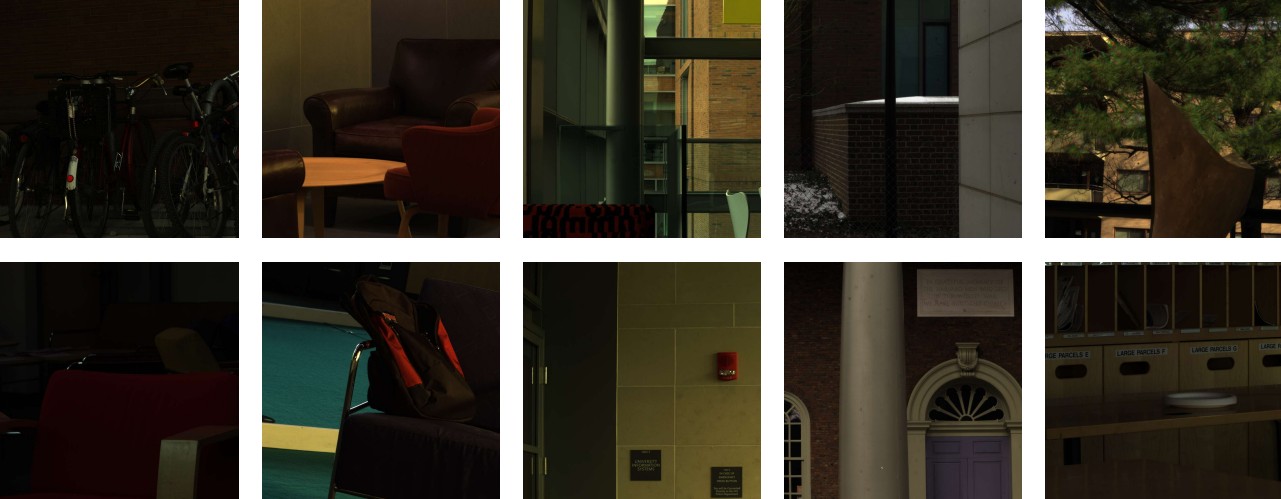

*Figure 9.* The 10 testing images from the Harvard dataset. A pseudo-color representation is used combining the 22th, the 15th, and the 7th bands.

### C.2. Quality Metrics

We compare our method with other methods using different image quality metrics to validate the image fusion capability of our model, including the Spectral Angle Mapper (SAM) (Yuhas et al., 1992), the Erreur Relative Globale Adimensionnelle de Synthèse (ERGAS) (Wald, 2002), the Peak Signal-to-Noise Ratio (PSNR) (Hore & Ziou, 2010).

**PSNR** evaluates the spatial quality of each band in the reconstructed HR-HSI. It is calculated as follows:

$$\text{PSNR}(R, O) = \frac{1}{C} \sum_{i=1}^{C} \text{PSNR}(R^i, O^i), \tag{23}$$

Here, $R^i \in \mathbb{R}^{H \times W}$ and $O^i \in \mathbb{R}^{H \times W}$ represent the $i$-th band of $R \in \mathbb{R}^{H \times W \times C}$ and $O \in \mathbb{R}^{H \times W \times C}$, respectively. The PSNR function is defined as:

$$\text{PSNR}(R^i, O^i) = 20 \cdot \log_{10}\left(\frac{DataRange}{\sqrt{\text{MSE}(R^i, O^i)}}\right), \tag{24}$$

where MSE (Mean Square Error) between $R^i$ and $O^i$, and $DataRange = 1.0$ since the values of $R$ and $O$ lie in $[0, 1]$.

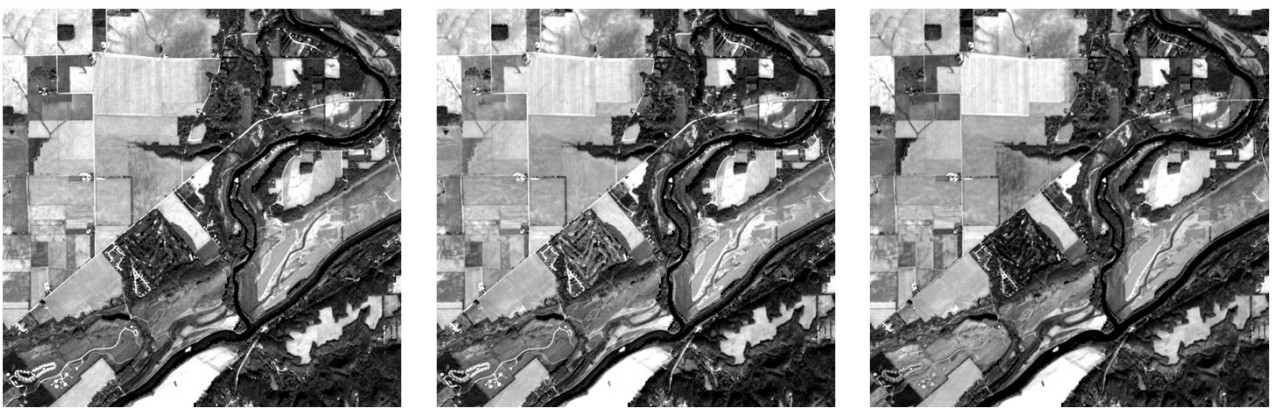

*Figure 10.* Band-wise visualizations of the Sentinel-2 image.

Channels 0~5

Channels 6~11

Channels 12~17

Channels 18~23

Channels 24~29

Channel 30

*Figure 11.* Band-wise visualizations of the Hyperion image.

*Table 8.* Notation used in Quality Metrics.

| Symbol | Meaning |
|---|---|
| $R \in \mathbb{R}^{H \times W \times C}$ | Reference (ground-truth) HR-HSI cube |
| $O \in \mathbb{R}^{H \times W \times C}$ | Reconstructed/predicted HR-HSI cube |
| $H, W$ | Spatial height and width of the HSI |
| $C$ | Number of spectral bands |
| $R^i \in \mathbb{R}^{H \times W}$ | The $i$-th band (spectral channel) of $R$ |
| $O^i \in \mathbb{R}^{H \times W}$ | The $i$-th band (spectral channel) of $O$ |
| $R_i \in \mathbb{R}^{1 \times C}$ | Spectrum vector at the $i$-th pixel location of $R$ (after flattening $HW$ pixels) |
| $O_i \in \mathbb{R}^{1 \times C}$ | Spectrum vector at the $i$-th pixel location of $O$ (after flattening $HW$ pixels) |
| $i$ | Band index in PSNR/ERGAS ($i = 1, \ldots, C$) or pixel index in SAM ($i = 1, \ldots, HW$) |
| $\text{MSE}(A, B)$ | Mean squared error between two images/tensors $A$ and $B$ (same shape) |
| $\mu_{R^i}$ | Mean value of the $i$-th reference band $R^i$ |
| $\lVert \cdot \rVert_2$ | $\ell_2$ (Euclidean) norm |
| $(\cdot)^T$ | Transpose operator |
| $\log_{10}(\cdot)$ | Base-10 logarithm |
| $\cos^{-1}(\cdot)$ | Arccosine function (SAM angle) |
| $DataRange$ | Dynamic range of data values ($DataRange = 1.0$ for $[0, 1]$ normalization) |
| $\epsilon$ | Small threshold for numerical stability in SAM ($\epsilon = 10^{-8}$) |
| $c$ | Scaling factor (ground sample distance ratio) used in ERGAS (fixed to $c = 4$ in this work) |

**SAM** measures the spectral distortion of each hyperspectral pixel in the reconstructed HR-HSI. It is given by:

$$\text{SAM}(R, O) = \frac{1}{HW} \sum_{i=1}^{HW} \cos^{-1} \left( \frac{R_i^T O_i}{\lVert R_i \rVert_2 \lVert O_i \rVert_2} \right), \tag{25}$$

where $cos^{-1}$ is the arccosine function, $R_i \in \mathbb{R}^{B \times 1}$ and $O_i \in \mathbb{R}^{B \times 1}$ are the spectra of the $i$-th pixel of $R$ and $O$, respectively, $\lVert \cdot \rVert_2$ is the $\ell_2$ norm, and $\cdot^T$ denotes the transpose. We report SAM in degrees. Pixels with $\lVert R_i \rVert_2 \lVert O_i \rVert_2 \leq \epsilon$ are excluded from averaging for numerical stability, where $\epsilon = 10^{-8}$.

**ERGAS** measures the global statistical quality of the reconstructed HR-HSI, taking into account the ratio of the ground sample distances between HR-MSI and LR-HSI. It is formulated as:

$$\text{ERGAS}(R, O) = \frac{100}{c} \sqrt{\frac{1}{B} \sum_{i=1}^{B} \frac{\text{MSE}(R^i, O^i)}{\mu_{R^i}^2}}, \tag{26}$$

where $c$ is the scaling factor, and $\mu_{R^i}^2$ is the square of the mean value of $R$. In particular, we fix $c = 4$. Under this setting, we can compare the performance of the same algorithm across different scales. If you need to compute the metric with the correct value for a specific setup, please adjust $c$ accordingly.

Higher PSNR values indicate better performance, while lower SAM and ERGAS values signify higher quality of the reconstructed HR-HSI. Ideally, PSNR should be infinite, SAM and ERGAS should be zero.

We will release our metric computation code on GitHub in a timely manner to facilitate reproducibility for other researchers.

### C.3. Comparison Methods

We follow the authors' official codebases to reproduce the compared methods:

- DCT (Ma et al., 2024): `https://github.com/qingma2016/DCTransformer`.

- LFormer (Hou et al., 2024): `https://github.com/coder-JMHou/LFormer`.

- MIMO (Fang et al., 2024): `https://github.com/Freelancefangjian/MIMO-SST`.

- FeINFN (Liang et al., 2024): `https://github.com/294coder/Efficient-MIF`.

- KNLConv (Ran et al., 2024): `https://github.com/Evangelion09/KNLNet`.

- SpecSolver (Li et al., 2025d): `https://github.com/weili419/SpecSolver`.

- Otias (Deng et al., 2025): `https://github.com/shangqideng/OTIAS`.

Please note that the FLOPs reported in Tables 1 and 2 are computed during training with a batch size of 32, whereas the PSNR, ERGAS, and SAM results are obtained using the hyperparameter settings provided in the original papers to ensure a fair comparison.

# D. More Results

## D.1. Statistical Significance Tests

*Table 9.* The average and standard deviation calculated for all the methods on the CAVE dataset simulating scaling factors of 4 and 8. The best results are in bold, second-best in underline. Params and FLOPs are reported in millions (M) and billions (G), respectively. $*$ indicates $p < 0.01$, and $\dagger$ indicates $p < 0.05$.

| Methods | Params | FLOPs | CAVE $\times 4$ | | | CAVE $\times 8$ | | |
|---|---|---|---|---|---|---|---|---|
| | | | PSNR $\uparrow$ | SAM $\downarrow$ | ERGAS $\downarrow$ | PSNR $\uparrow$ | SAM $\downarrow$ | ERGAS $\downarrow$ |
| Bicubic | - | - | 31.175±3.428 | 4.683±1.692 | 9.629±5.020 | 27.383±3.287 | 6.449±2.433 | 14.349±6.179 |
| DCT$_{2024}$ | 8.15 | 2457.43 | 50.196±3.359* | 2.486±0.799* | 1.198±0.738† | 47.441±4.532* | 3.164±0.824* | 1.791±1.360* |
| LFormer$_{2024}$ | 2.28 | 306.94 | 49.847±3.712* | 2.584±0.746* | 1.313±0.890* | 47.302±4.389* | 3.474±1.046* | 1.813±1.329* |
| MIMO$_{2024}$ | 4.98 | **49.08** | 50.101±2.997* | 2.767±0.875* | 1.237±0.701* | 47.442±3.781* | 3.985±1.356* | 1.758±1.120* |
| FeINFN$_{2024}$ | 3.17 | 382.72 | 50.358±3.547* | 2.537±0.791* | 1.210±0.816* | 47.476±4.364* | 3.565±1.157* | 1.755±1.350* |
| KNLConv$_{2024}$ | **1.73** | 114.04 | 48.570±4.809* | 2.790±0.903* | 1.557±1.006* | 46.150±4.317* | 3.639±1.161* | 1.989±1.318* |
| SpecSolver$_{2025}$ | 3.10 | 364.75 | 50.568±3.895* | 2.489±0.793* | 1.212±0.935* | 47.831±4.899* | 3.527±1.152* | 1.805±1.698* |
| Otias$_{2025}$ | 2.99 | 278.35 | 50.213±3.438* | 2.538±0.784* | 1.211±0.799* | 47.769±4.272* | 3.201±0.921* | 1.757±1.324* |
| Ours | 1.94 | 222.85 | **50.728±3.575** | **2.381±0.728** | **1.170±0.804** | **48.118±4.589** | **2.961±0.838** | **1.675±1.427** |
| *F*-statistic | | | 24.324* | 3.859* | 8.844* | 28.941* | 4.655* | 10.197* |

*Table 10.* The average and standard deviation calculated for all the methods on the Harvard dataset simulating scaling factors of 4 and 8. The best results are in bold, second-best in underline. Params and FLOPs are reported in millions (M) and billions (G), respectively. $*$ indicates $p < 0.01$, and $\dagger$ indicates $p < 0.05$.

| Methods | Params | FLOPs | Harvard $\times 4$ | | | Harvard $\times 8$ | | |
|---|---|---|---|---|---|---|---|---|
| | | | PSNR $\uparrow$ | SAM $\downarrow$ | ERGAS $\downarrow$ | PSNR $\uparrow$ | SAM $\downarrow$ | ERGAS $\downarrow$ |
| Bicubic | - | - | 37.173±3.955 | 2.645±0.700 | 5.445±2.389 | 34.051±3.978 | 3.186±0.883 | 7.587±3.198 |
| DCT$_{2024}$ | 8.15 | 2457.43 | 49.068±2.792* | 2.163±0.543† | 1.756±0.758* | 47.836±3.400* | 2.578±0.678* | 2.088±1.108* |
| LFormer$_{2024}$ | 2.28 | 306.94 | 48.938±2.947* | 2.182±0.567* | 1.706±0.642* | 46.651±3.279* | 2.742±0.741* | 2.305±1.133* |
| MIMO$_{2024}$ | 4.98 | **49.08** | 49.019±2.707* | 2.179±0.544* | 1.688±0.667* | 47.839±3.304* | 2.475±0.691* | 1.958±0.897* |
| FeINFN$_{2024}$ | 3.17 | 382.72 | 49.001±2.944* | 2.155±0.559* | 1.683±0.597* | 47.833±3.433* | 2.466±0.711* | 1.933±0.788* |
| KNLConv$_{2024}$ | **1.73** | 114.04 | 48.536±2.930* | 2.203±0.573* | 1.940±0.948* | 46.706±3.825* | 2.588±0.743* | 2.487±1.441* |
| SpecSolver$_{2025}$ | 3.10 | 364.75 | 48.919±2.827* | 2.169±0.565* | 1.771±0.737* | 47.652±3.306* | 2.533±0.710* | 2.052±0.964* |
| Otias$_{2025}$ | 2.99 | 278.35 | 49.046±2.905* | 2.151±0.561* | 1.728±0.657* | 47.848±3.309† | 2.459±0.680* | 1.907±0.725* |
| Ours | 1.94 | 222.85 | **49.182±2.924** | **2.144±0.550** | **1.658±0.609** | **47.957±3.340** | **2.438±0.671** | **1.879±0.732** |
| *F*-statistic | | | 29.684* | 4.307* | 13.746* | 32.257* | 5.690* | 15.437* |

We conduct paired two-sided $t$-tests between our method and other strong baselines in the main text, and report the corresponding $p$-values. Furthermore, to assess the overall differences among the eight algorithms (excluding bicubic), we compute the $F$-statistic based on the inter-group variance (measuring how far the mean of each method deviates from the global mean) and the intra-group variance (measuring how far each sample deviates from its method mean). The resulting $F$-value for PSNR on the CAVE dataset at the $\times 4$ scaling factor reaches 24.324 (with a corresponding $p$-value $< 0.01$), indicating that the performance differences across methods are substantially larger than the variability within each method.

To account for multiple comparisons, we apply a Bonferroni correction over the seven pairwise tests between our method and the other methods for each metric. With a family-wise significance level of $\alpha = 0.05$, the corrected threshold is $\alpha/7 \approx 7.1 \times 10^{-3}$. All reported $p$-values are well below this threshold, indicating that the improvements of our method remain statistically significant after correction. Taken together, these results provide strong evidence that our approach not only achieves superior numerical performance but also delivers statistically significant improvements over competing methods, demonstrating high robustness.

## D.2. Results on the Real-world Dataset

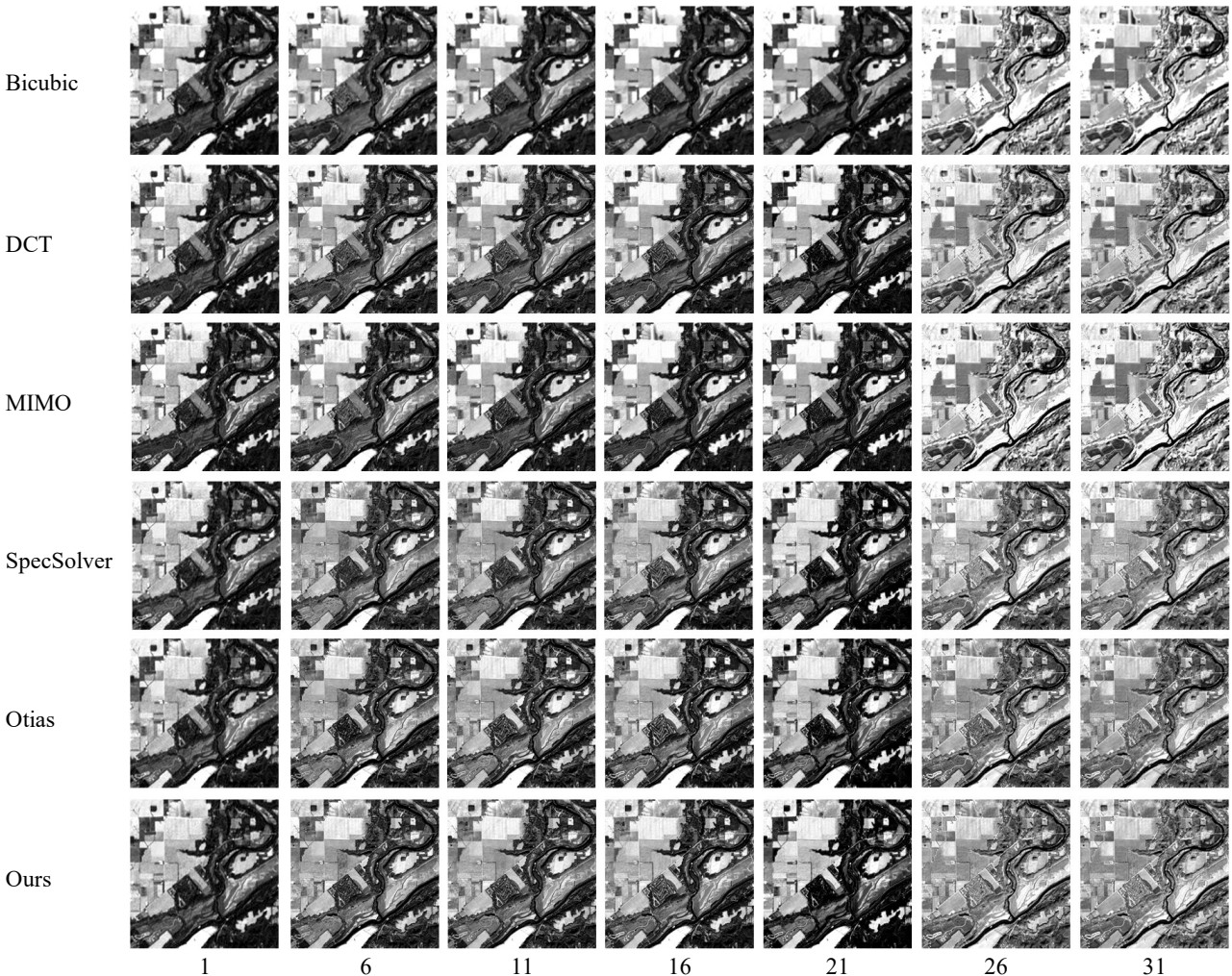

*Figure 12.* For the reconstruction results on the real-world dataset, we visualize the selected bands ([1, 6, 11, 16, 21, 26, 31]).

Due to space limitations, we provide additional quantitative and qualitative results in the appendix. As reported in Table 11, we evaluate $D_\lambda$, $D_s$, and QNR. Our method achieves the best $D_\lambda$ score and the second-best $D_s$ score, indicating the strongest spectral reconstruction quality and competitive spatial reconstruction performance. Moreover, it attains the highest overall QNR, which we attribute to the structured control introduced by the Trigonometric Basis Solver. In addition, Fig. 12 presents more visualization results, including those obtained with bicubic interpolation. At first glance, all methods produce reasonable reconstructions; however, after zooming in, our method preserves finer details more clearly and yields sharper, better-defined object boundaries. These observations suggest that our approach not only achieves superior quantitative scores but also delivers the best visual quality.

*Table 11.* Quality indices on the real dataset. Lower is better for $D_\lambda$ and $D_s$, while higher is better for QNR.

| Methods | $D_\lambda \downarrow$ | $D_s \downarrow$ | QNR $\uparrow$ |
|---|---|---|---|
| DCT$_{2024}$ | 0.18724 | 0.31133 | 0.55973 |
| LFormer$_{2024}$ | 0.22144 | 0.25031 | 0.58368 |
| MIMO$_{2024}$ | 0.17571 | 0.26019 | 0.60982 |
| FeINFN$_{2024}$ | 0.21737 | 0.26265 | 0.57708 |
| KNLConv$_{2024}$ | 0.25023 | 0.39568 | 0.45311 |
| SpecSolver$_{2025}$ | 0.18222 | **0.24395** | 0.61828 |
| Otias$_{2025}$ | 0.19103 | 0.26345 | 0.59585 |
| Ours | **0.16994** | 0.24441 | **0.62718** |

## D.3. Hyperparameter Ablation Study

From Table 12, we observe that the model performance is primarily governed by the coupled effect of $n_\text{token}$ and $n_\text{basis}$, whereas the Width and Patch size mainly determine the boundary of the performance–complexity trade-off.

First, with Width fixed to 64 and Patch fixed to 4, increasing $n_\text{basis}$ yields consistent gains. When $n_\text{token} = 6$, raising $n_\text{basis}$ from 22 to 28 improves PSNR from 48.901 to 49.166, while reducing SAM from 2.223 to 2.142 and ERGAS from 1.752 to 1.632. This indicates that a richer basis enhances the expressive power for spatial–spectral reconstruction. Notably, the improvement comes with only a slight increase in FLOPs (approximately 219G→225G) and almost unchanged parameters (around 1.94M), suggesting an effective direction that increases computation with negligible parameter growth.

Second, increasing $n_\text{token}$ does not lead to monotonic improvement; instead, the performance saturates and may even degrade slightly. For example, with $n_\text{basis} = 24$, PSNR peaks at 49.182 when $n_\text{token} = 8$, and further increasing $n_\text{token}$ to 10/12/14 does not improve results (49.146/49.156/49.137). SAM and ERGAS also remain largely unchanged with minor fluctuations. This suggests that an overly large token set may introduce redundant representations or increase optimization difficulty, diminishing marginal returns. Under this setting, $n_\text{token} = 8$ offers a better cost–effectiveness point. Across all configurations, the best PSNR among the listed combinations is achieved at (Width=64, Patch=4, $n_\text{token} = 6$, $n_\text{basis} = 28$) with 49.166. Meanwhile, (64, 4, 8, 26) attains a very comparable PSNR of 49.188 under similar computational cost, implying a degree of interchangeability in allocating capacity between tokens and basis functions.

Finally, the ablations on Width and patch size reveal a typical "compute-for-performance" pattern. Increasing Width brings small but consistent improvements. For instance, with Patch=2, $n_\text{token} = 8$, and $n_\text{basis} = 24$, increasing Width from 48 to 96 raises PSNR from 48.620 to 49.150, decreases SAM from 2.310 to 2.140, and decreases ERGAS from 1.812 to 1.620, but at the cost of noticeably higher parameters and FLOPs (1.709M/205.8G → 2.590M/269.4G). In contrast, varying the patch size (2/4/8) has a relatively minor effect: under the same Width (e.g., 64 or 96), the differences in PSNR, SAM, and ERGAS remain limited, suggesting that patch granularity is not the primary bottleneck for this task. One possible explanation is that the Hierarchical Projection Network already provides rich multi-scale information for the SSF task, thereby reducing the sensitivity to the choice of patch size. Instead, model capacity (Width) and representational allocation ($n_\text{token}/n_\text{basis}$) play a more critical role.

Overall, Table 12 supports the following conclusions: (1) increasing $n_\text{basis}$ generally yields stable benefits; (2) $n_\text{token}$ has an optimal range (around 8), beyond which gains are marginal; (3) larger Width improves performance but significantly increases computational and parameter costs; and (4) patch size has a secondary influence and mainly serves as an efficiency knob in practice.

Considering the overall performance together with the parameter count and computational cost, we set (Width=64, Patch=4, $n_\text{token} = 8$, $n_\text{basis} = 24$) as the default hyperparameters for the CAVE and Harvard datasets throughout this paper.

## D.4. Structural Ablation Study

In the main text, we perform feature fusion in the latent space using a trigonometric-basis expansion. To verify its effectiveness, we replace this module with alternative designs—namely an MLP, convolution, and self-attention, and report the corresponding results, as shown in Table 13. Across different design choices, Trigonometric-Basis achieves the best

*Table 12.* Ablation study on width, patch size, token number, and basis number. The best results are in bold.

| Width | Patch | $n_{\text{token}}$ | $n_{\text{basis}}$ | PSNR ↑ | SAM ↓ | ERGAS ↓ | Params (M) ↓ / FLOPs (G) ↓ |
|---|---|---|---|---|---|---|---|
| 64 | 4 | 6 | 22 | 48.901 | 2.223 | 1.752 | 1.942 / 219.329 |
| 64 | 4 | 6 | 24 | 49.026 | 2.180 | 1.700 | 1.942 / 221.317 |
| 64 | 4 | 6 | 26 | 49.102 | 2.146 | 1.643 | 1.943 / 223.278 |
| 64 | 4 | 6 | 28 | 49.166 | 2.142 | 1.632 | 1.944 / 225.496 |
| 64 | 4 | 8 | 22 | 49.073 | 2.151 | 1.686 | 1.942 / 220.730 |
| 64 | 4 | 8 | 24 | 49.182 | 2.144 | 1.658 | 1.943 / 222.850 |
| 64 | 4 | 8 | 26 | 49.188 | 2.138 | 1.645 | 1.943 / 224.269 |
| 64 | 4 | 8 | 28 | 49.179 | 2.143 | 1.671 | 1.944 / 226.424 |
| 64 | 4 | 8 | 32 | 49.181 | 2.151 | 1.656 | 1.946 / 229.336 |
| 64 | 4 | 10 | 22 | 49.151 | 2.141 | 1.656 | 1.942 / 221.734 |
| 64 | 4 | 10 | 24 | 49.146 | 2.142 | 1.661 | 1.943 / 223.867 |
| 64 | 4 | 10 | 26 | 49.126 | 2.153 | 1.632 | 1.944 / 225.448 |
| 64 | 4 | 10 | 28 | 49.084 | 2.156 | 1.650 | 1.945 / 227.703 |
| 64 | 4 | 12 | 22 | 49.109 | 2.153 | 1.664 | 1.943 / 222.268 |
| 64 | 4 | 12 | 24 | 49.156 | 2.146 | 1.655 | 1.943 / 224.323 |
| 64 | 4 | 12 | 26 | 49.179 | 2.142 | 1.639 | 1.944 / 226.334 |
| 64 | 4 | 12 | 28 | 49.155 | 2.141 | 1.663 | 1.945 / 228.897 |
| 64 | 4 | 14 | 22 | 49.099 | 2.154 | 1.673 | 1.943 / 223.654 |
| 64 | 4 | 14 | 24 | 49.137 | 2.148 | 1.665 | 1.944 / 225.771 |
| 64 | 4 | 14 | 26 | 49.157 | 2.153 | 1.650 | 1.945 / 227.972 |
| 64 | 4 | 14 | 28 | 49.115 | 2.149 | 1.685 | 1.945 / 229.659 |
| 48 | 2 | 8 | 24 | 48.620 | 2.310 | 1.812 | 1.709 / 205.751 |
| 64 | 2 | 8 | 24 | 48.940 | 2.190 | 1.695 | 1.943 / 222.848 |
| 80 | 2 | 8 | 24 | 49.080 | 2.155 | 1.645 | 2.236 / 244.060 |
| 96 | 2 | 8 | 24 | 49.150 | 2.140 | 1.620 | 2.590 / 269.387 |
| 48 | 4 | 8 | 24 | 48.700 | 2.290 | 1.785 | 1.709 / 205.753 |
| 64 | 4 | 8 | 24 | 49.182 | 2.144 | 1.658 | 1.943 / 222.850 |
| 80 | 4 | 8 | 24 | 49.160 | 2.150 | 1.630 | 2.236 / 244.062 |
| 96 | 4 | 8 | 24 | 49.180 | 2.145 | 1.615 | 2.590 / 269.389 |
| 48 | 8 | 8 | 24 | 48.640 | 2.305 | 1.805 | 1.709 / 205.754 |
| 64 | 8 | 8 | 24 | 48.980 | 2.185 | 1.690 | 1.943 / 222.851 |
| 80 | 8 | 8 | 24 | 49.100 | 2.160 | 1.650 | 2.236 / 244.063 |
| 96 | 8 | 8 | 24 | 49.130 | 2.155 | 1.630 | 2.590 / 269.390 |

*Table 13.* Comparison of different designs.

| Designs | PSNR ↑ | SAM ↓ | ERGAS ↓ | Params (M) ↓ / FLOPs (G) ↓ | |
|---|---|---|---|---|---|
| MLP | 47.169 | 2.866 | 1.931 | 1.992 | 254.260 |
| Convolution | 48.672 | 2.531 | 1.882 | 1.956 | 223.008 |
| Self-Attention | 48.865 | 2.258 | 1.673 | 1.993 | 347.076 |
| Trigonometric-Basis | 49.182 | 2.144 | 1.658 | 1.943 | 222.850 |

performance on all three metrics (PSNR 49.182 / SAM 2.144 / ERGAS 1.658). Compared with the MLP variant, it improves PSNR by 2.013 dB and reduces SAM and ERGAS by 0.722 and 0.273, respectively. It also brings consistent gains over the convolutional alternative (PSNR +0.510 dB, SAM -0.387, ERGAS -0.224). Notably, these improvements are not attributed to a larger model size, as all variants have roughly 2M parameters. In terms of computational cost, Trigonometric-Basis requires 222.850G FLOPs, almost identical to the convolutional design (223.008G), while offering a clearly better accuracy–efficiency trade-off than self-attention, although self-attention attains the second-best accuracy (PSNR 48.865 / SAM 2.258 / ERGAS 1.673), it incurs substantially higher computation (347.076G FLOPs). Overall, Trigonometric-Basis delivers the strongest reconstruction quality while maintaining low complexity, demonstrating a superior accuracy–efficiency balance.

Thanks to the Trigonometric-Basis Solver, we can explicitly control the amount of high-frequency information injected from the input, thereby avoiding excessively sharp high-frequency signals. As shown in the Fig. 13, the trigonometric-basis expansion yields a noticeably smoother and more stable response in the high-frequency range, whereas Self-Attention designs tend to produce spiky peaks at high frequencies.

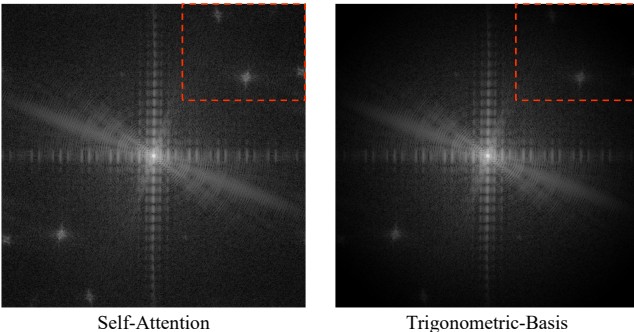

Self-Attention     Trigonometric-Basis

*Figure 13.* Fourier amplitude analysis of reconstructions from Self-Attention and Trigonometric-Basis designs.

