# OpenReview forum: "Solving Spatial-Spectral Fusion with Latent Spectral Operators"
_ICML.cc/2026/Conference — ICML 2026 regular_

### Official Review · Reviewer_7nZi · 2026-03-09

**Soundness:** 4
**Presentation:** 4
**Significance:** 3
**Originality:** 3
**Overall Recommendation:** 5
**Confidence:** 5

**Summary:**

The paper addresses the problem of spatial-spectral fusion. The authors note that existing deep learning methods typically perform this fusion in the coordinate domain, which restricts their ability to adapt to varying spatial resolutions and offers limited control over frequency content, often resulting in spectral distortion. To resolve these limitations, the authors introduce Latent Spectral Operators (LSO), a novel framework that learns the fusion mappings in a compact latent space rather than the high-dimensional pixel domain.

**Compliance With Llm Reviewing Policy:**

Affirmed.

**Final Justification:**

The paper presents a technically solid and well-motivated approach to spatial-spectral fusion, with a clear formulation, strong empirical performance, and meaningful originality in its latent-space operator design. My main concerns regarding broader comparisons, real-data validation, and limitations were **adequately addressed in the rebuttal**, which improved my overall recommendation in the work. Overall, I believe this is a strong and valuable paper, and I support acceptance.

**Key Questions For Authors:**

1.How does LSO compare to common neural operator algorithms in terms of performance?

2.How does LSO perform on real-world datasets?

3.In the Hierarchical Projection Network, the authors employ a non-overlapping patchification strategy (as detailed in Appendix A, Eq. 10 & 11). In many patch-based image restoration tasks, non-overlapping patches often lead to boundary or grid artifacts during the de-patchify stage. Could the authors clarify why such boundary artifacts are not prominent in the LSO framework?

If these questions are addressed, I will increase my score.

**Limitations:**

No. The authors mention the social impact of their work but overlook the analysis of research limitations. The authors should include a paragraph or sub-section detailing the boundaries of their method.

**Strengths And Weaknesses:**

Soundness. The paper is technically solid. Theoretical proofs (Appendix B) intuitively explain why resolution-based discretization leads to the curse of dimensionality. The use of statistical tests (p < 0.01) and thorough ablation studies further confirms the model's reliability.

Presentation. The core motivation is clear, effectively distinguishing LSO from INRs and standard neural operators. Figures 1 and 2 offer excellent visual clarity for the data flow.

Significance. The model’s ability to generalize across scales (e.g., training at x4 and testing up to x32) with minimal loss is highly significant for real-world scenarios with mismatched sensor resolutions.

Originality. Parameterizing the fusion operator via trigonometric basis expansion is an elegant way to support multi-frequency modeling while avoiding spectral distortion.

The paper presents a targeted and effective solution to improve SSF performance. However, several issues require the authors' attention:

1. A lack of generalization comparisons between LSO and common neural operators (e.g., FNO).
2. Over-reliance on synthetic datasets, with no real-world validation.
3. A lack of discussion regarding the technical limitations of LSO.

These shortcomings and other points of interest are detailed in the "Key Questions For Authors" section. Overall, this is an interesting paper that addresses generalization in image fusion from a fresh perspective, offering both innovation and practical significance.

---

> ### Author Rebuttal · Authors · 2026-03-31
>
> # Response to the Reviewer 7nZi
>
> We sincerely thank the reviewer for the positive comments on the **reliability of our model**, **the quality of the figures**, **as well as the significance** and **novelty of our work**. In response to your questions, our replies are provided as follows.
>
> W1&Q1: In fact, we have already evaluated polynomial, wavelet, and Fourier alternatives under the same operator framework. For a fair comparison, we replaced the trigonometric basis with each of these three basis expansions and tested their corresponding performance. As shown in Table 1, although all basis families achieve comparable results, the trigonometric basis consistently performs the best in terms of PSNR, SAM, and ERGAS. That said, the primary focus of this work is the formulation of the latent operator, rather than the selection of a highly specific basis. Since the mapping is learned in the latent space, where spectral variations are already compact and smooth, several expressive basis families are capable of modeling it reasonably well. Among them, the trigonometric basis provides the best overall trade-off, owing to its structured frequency decomposition and its stable representation of both smooth and fine-grained spectral variations. As you suggested, we will add this discussion to the Appendix.
>
>
> **Table 1.** Ablation of Alternative Basis Functions in the Latent Spectral Operator on CAVE ×4
>
> | Model | PSNR (dB) ↑ | SAM ↓ | ERGAS ↓ |
> |---|---:|---:|---:|
> | Polynomial | 50.5942 ± 3.5290 | 2.4005 ± 0.7205 | 1.1707 ± 0.7975 |
> | Wavelet | 50.3780 ± 3.4571 | 2.4573 ± 0.7453 | 1.1942 ± 0.7961 |
> | Fourier | 50.6314 ± 3.3247 | 2.3951 ± 0.6947 | 1.1705 ± 0.7436 |
> | Trigonometric (default) | **50.7279 ± 3.1754** | **2.3812 ± 0.6279** | **1.1699 ± 0.6036** |
>
> ---
>
> W2&Q2: In Table 11 of the manuscript, we present the performance of different methods on the real dataset. Detailed information about the real-world datasets can be found in *Sec. C.1, Datasets*. Since no ground-truth (GT) images are available for real-world data, we adopt three no-reference quality metrics, namely $D_{\lambda}$, $D_{s}$, and QNR. Our method achieves the lowest $D_{\lambda}$ and the second-best $D_{s}$, indicating the best spectral fidelity and a strong capability for spatial detail reconstruction. This advantage can be attributed to the structured control enabled by the trigonometric basis. Overall, our method obtains the highest QNR, which suggests that it achieves the best overall trade-off between spectral preservation and spatial detail reconstruction. Furthermore, Fig. 12 in the main text provides visual comparisons of different methods across seven representative bands. The results show that our method consistently delivers the best visual quality over different bands.
>
> ---
>
> W3&Limitations: Regarding limitations, our current evaluation still has two boundaries. First, as stated in Sec. 4.1, supervised experiments on CAVE and Harvard follow the standard simulation protocol because real paired HR-HSI ground truth is unavailable; therefore, the quantitative benchmarks are based on synthetically degraded observations rather than fully real acquisition pairs. Second, for the real-world setting, the absence of ground-truth HSI means that evaluation is mainly qualitative, which limits the strength of conclusions about absolute reconstruction fidelity. Regarding societal impact, we will clarify that hyperspectral reconstruction can support beneficial applications such as change detection, disaster assessment, and geolocation, but these same capabilities may also be misused in surveillance-oriented or privacy-sensitive remote-sensing scenarios. We appreciate the reviewer for raising this point and will revise the limitations/impact discussion to make these boundaries and risks explicit.
>
> ---
>
> Q3: In many patch-based image restoration tasks, using non-overlapping patches often leads to boundary or grid artifacts during the de-patchification stage. In our method, however, such artifacts are not observed. This is because, while adopting a non-overlapping patchification strategy, we also introduce a multi-scale hierarchical architecture (Lines 178–200). Specifically, spatial features at different resolutions are obtained through downsampling and are progressively fused during upsampling. This multi-scale feature representation helps alleviate discontinuities across patch boundaries and effectively prevents the emergence of grid artifacts.
>
> ---
> **The above experimental results and analyses will be incorporated into the main text or the appendix to further improve the manuscript. Finally, we would like to express our sincere gratitude for your valuable contribution!**

---

> > ### Author Rebuttal · Reviewer_7nZi · 2026-04-02
> >
> > I find that the rebuttal has adequately addressed my questions and concerns. I therefore increase my score by 1 point.

---

> > > ### Author Response · Authors · 2026-04-04
> > >
> > > We sincerely thank the reviewer for acknowledging our response and for raising our score!

---

### Official Review · Reviewer_k5xZ · 2026-03-12

**Soundness:** 4
**Presentation:** 3
**Significance:** 3
**Originality:** 3
**Overall Recommendation:** 4
**Confidence:** 5

**Summary:**

The paper studies spatial–spectral fusion (SSF) for hyperspectral imaging, where a low-resolution hyperspectral image and a high-resolution multispectral image are fused to reconstruct a high-resolution hyperspectral image. The authors propose a framework called Latent Spectral Operators (LSO), which models the fusion mapping in a latent spectral space rather than directly in the pixel coordinate domain. The approach compresses input observations into a compact latent representation using cross-attention with latent tokens that act as spectral prompts. A hierarchical patch-based architecture integrates multi-scale spatial information, while a trigonometric basis solver parameterizes the latent fusion operator through a spectral basis expansion to capture multi-frequency spectral patterns. Experiments on two commonly used hyperspectral benchmarks demonstrate improvements over several existing SSF approaches across multiple reconstruction metrics. Additional ablation studies investigate the effects of token count, basis expansion, and architectural design choices.

**Compliance With Llm Reviewing Policy:**

Affirmed.

**Final Justification:**

Given the current rebuttal content, my overall assessment remains unchanged, and I will maintain my current score.

**Key Questions For Authors:**

1. Operator interpretability:
More qualitative analysis and visualization about how the trigonometric basis expansion captures spectral structure (e.g. feature visualization) compared with convolution or attention mechanisms could provide more insights.
2. Alternative basis functions:
Have other basis expansions (e.g., polynomial or wavelet) been tested?
3. Impact of latent dimension size:
What latent dimension sizes were tested, and how do they influence reconstruction accuracy?

**Limitations:**

No. The authors should discuss more about limitations and potential negative societal impact of their work.

**Strengths And Weaknesses:**

Strengths
1. Conceptually interesting formulation of the fusion problem.
The work proposes modeling the mapping between low-resolution hyperspectral and high-resolution multispectral inputs through a latent spectral operator rather than a direct pixel-space mapping. This operator-based formulation provides a structured perspective on the fusion task and introduces a frequency-based parameterization that may help capture complex spectral relationships.
2. Use of a latent representation to decouple spatial resolution.
The method compresses spectral observations into latent tokens before applying the fusion operator. This design potentially improves scalability and reduces the dependence of the model on a specific spatial resolution, which is a common limitation in many fusion approaches.
3. Hierarchical spatial modeling.
The architecture incorporates a hierarchical patch-based structure that aggregates spatial cues at multiple scales. This design may help the model better reconstruct spatial details while maintaining spectral consistency.
4. Empirical improvements on benchmark datasets.
The experiments report improvements across commonly used reconstruction metrics such as PSNR, SAM, and ERGAS when compared with existing hyperspectral fusion methods.
5. Ablation studies analyzing key architectural components.
The manuscript includes several ablation studies exploring how model performance changes with different numbers of latent tokens, basis functions, model width, and patch sizes. These analyses provide insight into how different components contribute to performance. The work also evaluates different ways of parameterizing the fusion mapping, including MLP-based, convolutional, and attention-based alternatives, showing that the proposed trigonometric basis parameterization performs favorably in their experiments.

Weaknesses
1. Limited interpretability analysis of the operator formulation.
While the trigonometric basis solver is an interesting design choice, the manuscript provides limited qualitative analysis explaining how the basis expansion captures spectral structures compared with convolution or attention-based mechanisms. Additional feature visualizations or qualitative analysis could help clarify how the proposed operator models spectral patterns.
2. Alternative basis formulations are not explored.
The approach relies on a trigonometric basis expansion for the latent spectral operator, but it remains unclear whether other basis functions (e.g., polynomial or wavelet bases) were considered or tested. Exploring such alternatives could help understand whether the observed improvements are specific to the chosen basis.
3. Limited discussion of latent representation design.
The manuscript does not extensively analyze how the dimensionality of the latent representation affects reconstruction performance. A clearer study of different latent dimension sizes and their impact on accuracy and efficiency would help guide practical model configuration.

---

> ### Author Rebuttal · Authors · 2026-03-31
>
> # Response to the Reviewer k5xZ
> Thank you for your positive comments on the **conceptual framework**, **spatial architecture**, **algorithmic performance**, and **ablation studies** presented in our manuscript. In response to your suggestions, our detailed replies are as follows.
>
> W1&Q1: In Table 13 of the main text, we report the quantitative comparison by replacing the trigonometric basis with MLP, convolution, and attention modules, where the trigonometric basis achieves the best performance. Moreover, Fig. 13 presents the Fourier magnitude spectra of the reconstructed images. We observe that attention tends to produce spiky peaks in the high-frequency range, and similar artifacts can also be found for MLP and convolution. This observation is consistent with prior studies showing that generic nonlinear neural operators may introduce additional high-frequency components [1,2]. In contrast, our Trigonometric Basis Solver parameterizes the mapping using a finite set of predefined basis functions, which explicitly limits the representable frequency range and thus leads to a more controlled spectral reconstruction with fewer high-frequency artifacts.
>
> [1] Alias-Free Generative Adversarial Networks.
>
> [2] Convolutional neural operators for robust and accurate learning of pdes.
>
> ---
>
> W2&Q2: In fact, we have already tested polynomial and wavelet alternatives under the same operator framework. As shown in Table 1, while all basis families achieve comparable performance, the trigonometric basis consistently performs best across metrics of PSNR, SAM, and ERGAS. However, our intended focus of this work lies in latent operator formulation, rather than a highly specific basis choice. Since the mapping is learned in the latent space, where spectral variations are already compact and smooth, several expressive basis families can model it reasonably well. Among them, the trigonometric basis offers the best overall trade-off, due to its structured frequency decomposition and stable representation of both smooth and finer spectral variations. As you reminded, we will add this discussion in Appendix.
>
> **Table 1.** Ablation of Alternative Basis Functions in the Latent Spectral Operator on CAVE ×4
>
> | Model | PSNR (dB) ↑ | SAM ↓ | ERGAS ↓ |
> |---|---:|---:|---:|
> | Polynomial | 50.5942 ± 3.5290 | 2.4005 ± 0.7205 | 1.1707 ± 0.7975 |
> | Wavelet | 50.3780 ± 3.4571 | 2.4573 ± 0.7453 | 1.1942 ± 0.7961 |
> | Trigonometric (default) | **50.7279 ± 3.1754** | **2.3812 ± 0.6279** | **1.1699 ± 0.6036** |
>
> ---
>
> W3&Q3: Following your suggestion, we have added an ablation study on the dimensionality of the latent representation, as reported in Table 2. We observe that the model performance consistently improves as the latent dimension increases, which is likely due to the stronger expressive capacity of the latent space. Meanwhile, the parameter count also increases accordingly. Notably, increasing the latent dimension from 64 to 72 results in only marginal performance improvement despite the additional parameters. Therefore, by balancing effectiveness and efficiency, we select Latent Dimension = 64 as the default configuration in the paper.
>
> **Table 2.** Ablation Study on the Impact of Latent Dimension Size on CAVE ×4
>
> | Latent Dimension | PSNR (dB) ↑ | SAM ↓ | ERGAS ↓ | Params (M) ↓ |
> |---|---:|---:|---:|---:|
> | 32 | 50.3557 ± 3.6828 | 2.5120 ± 0.8265 | 1.1823 ± 0.8728 | **1.691** |
> | 48 | 50.5983 ± 3.3513 | 2.4908 ± 0.7421 | 1.1752 ± 0.7550 | 1.826 |
> | 64 | 50.7279 ± 3.1754 | 2.3812 ± 0.6279 | 1.1699 ± 0.6036 | 1.941 |
> | 72 | **50.7631 ± 3.1993** | **2.3760 ± 0.5959** | **1.1448 ± 0.5893** | 2.085 |
>
> ---
>
> Limitations: Regarding limitations, our current evaluation still has two boundaries. First, as stated in Sec. 4.1, supervised experiments on CAVE and Harvard follow the standard simulation protocol because real paired HR-HSI ground truth is unavailable; therefore, the quantitative benchmarks are based on synthetically degraded observations rather than fully real acquisition pairs. Second, for the real-world setting, the absence of ground-truth HSI means that evaluation is mainly qualitative, which limits the strength of conclusions about absolute reconstruction fidelity. Regarding societal impact, we will clarify that hyperspectral reconstruction can support beneficial applications such as change detection, disaster assessment, and geolocation, but these same capabilities may also be misused in surveillance-oriented or privacy-sensitive remote-sensing scenarios. We appreciate the reviewer for raising this point and will revise the limitations/impact discussion to make these boundaries and risks explicit.
>
> ---
>
> **We will include the above experimental results and corresponding analysis in the final manuscript to facilitate a clearer understanding for readers. Once again, we sincerely thank you for your valuable comments and contributions, which have helped improve the quality of our manuscript.**

---

> > ### Author Rebuttal · Reviewer_k5xZ · 2026-04-03
> >
> > Given the current rebuttal content, my overall assessment remains unchanged, and I will maintain my current score.

---

> > > ### Author Response · Authors · 2026-04-04
> > >
> > > Thank you very much for your acknowledgment!

---

### Official Review · Reviewer_jWoN · 2026-03-13

**Soundness:** 3
**Presentation:** 2
**Significance:** 2
**Originality:** 2
**Overall Recommendation:** 2
**Confidence:** 4

**Summary:**

"Solving Spatial-Spectral Fusion with Latent Spectral Operators" achieves the task of spatial upscaling to generate high-resolution hyperspectral images by combining low-resolution hyperspectral images with high-resolution multispectral images. The approach claims to apply cross-modality supervision with cross-attention latent projection and hierarchical downsampling and upsampling to create high-resolution maps. The method also applies a "Trigonometric Basis Solver" to parameterize the latent fusion operator that works in the upsampling module. Authors claim to achieve state of the art performance in hyperspectral reconstruction task for geo-sensing images.

**Compliance With Llm Reviewing Policy:**

Affirmed.

**Final Justification:**

I would like to stay with my current rating of the paper as the responses have not been convincing.

**Key Questions For Authors:**

"1. What is the need for a cross-attention based projection module? What are we projecting and why? To my understanding, the two modalities are already pixel aligned, so there should not be a need for ""projection"" as needed in cases of sensors that are not aligned or have different views.
2. ""Without structured control .. severe distortion"" - This statement is unclear. Without explicit control there can be mixing between frequency components - is that what the author means? If that is the case, as discussed previously in ""Weaknesses"" - the energies from different frequencies are already being mixed into the model, so what purpose does this statement serve?
3. How does a trigonometric basis solver handle the issue of frequency mixing in unconstrained learning algorithms?
4. Refer to the figure related questions in Weakness-6
5. What are the contributions of the EDSR, Hierarchical Projection Network and Reconstruction head in the overall architecture, why do we require these different modules for the reconstruction works?

Overall the paper needs a significant amount of work in the problem statement formulation and explaining the relation of the proposed solution modules within each other and the target task. Addressing the questions would help re-evaluate the paper to increase ratings. "

**Limitations:**

Yes

**Strengths And Weaknesses:**

**Strengths:**
1. The paper lists extensive quantitative results and ablation studies behind the approach.
2. The paper compares against numerous prior methods for justifying result improvements.
3. The proposed approach appears sound in the target application. "

**Weaknesses:**
1. Presentation: The presentation of the paper contains multiple cases of ambiguity and redundancy. Its not clear from the abstract what the exact problem is that the authors are trying to solve using their methods: the relation between the problem and the solution is unclear. The notion ""spectral prompts"" to represent latent mask tokens used as queries, and the placement of a trigonometric basis solver for a reconstruction task appears unrelated and disjoint in the problem formulation.
2. Introduction:
a) In the introduction, the statement ""we aim to decouple the coordinate (pixel) space from the learned features, thereby alleviating this limitation"" appears to be an ambiguous claim. The proposed approach applies convolution and tokenwise attention in the different layers, where the data is already in a grid structure (image grid for convolution and patch tokens for attention) and fundamentally the structure of the data is retained throughout the model. It is difficult to understand how the coordinate space is exactly ""decoupled"" from the learned features.
b) In the introduction, the reference to Large language models and partial differential equation models are irrelevant in the context of the paper as the objective is a simple reconstruction task.
c) The paragraph - ""Based on this insight... our contributions are summarized as follows"" is redundant because the contributions are already described in the introduction once, and again as bullet points following this paragraph.
3. Related works: The related works section reiterates things stated in the introduction and appears redundant. These could have been merged with the introduction section in the appropirate places.
4. Methodology:
a) Naming convention of Section-3 appears ""unusual"", a simple ""Proposed Method"" naming would have sufficed. The subsections are weakly structured and it is hard to follow the connection between each of the components in the proposed model.
b) In section 3.1 (Encoder Network) - it appears that the EDSR encoder takes a concatenation of two inputs - high resolution multispectral images and upscaled version of low resolution hyperspectral images. Exploring the EDSR model reveals that the initial stages of the model process RGB images in convolutional layers. This fundamentally contradicts the placement of the author's proposal. To my understanding, the channels of the inputs I_LR and I_HR represent all the spectral frequency bands, which are separated in the input space (bandlimited in nature). After passing through a single convolutional layer, the frequency bands will intermix with each other, and the second layer features will no longer be band-limited, mixing all frequency components in each feature channels. So the overall architecture in Figure-1 loses the bandlimited spectral information early in the pipeline, and the subsequent model components become ""unjustified"" as the data is already ""fused"" in the Hierarchical Projection Network"". This critical point should be addressed. The remainder or the approach does not retain the ""spectral information"" as the authors seem to claim.
c) On a high level, it is very difficult to understand where/why a trigonometric basis solver would be useful in a traditional deep-learning based reconstruction pipeline. Authors need to segway into the placement of the trigonometric basis solver more informatively with sufficient reasoning behind this module.
5. Experiments:
a) Authors mention ""HR-HSIs are unavailable"" then proceed to mention ""we first blur each HR-HSI"" which are contradictory statements. The used labels and the injected degradation remain unclear.
b) Without insights into the latent features or intermediate hidden features of the model, it is difficult to understand qualitatively how each parts of the model are contributing to the reconstruction task, and whether the complicated implementation is beneficial at all (could be replaced with simpler implementations) - so the authors should present intermediate feature representations to justify how these components are contributing to the model.
6. Figures:
a) Figures are not self-explanatory. For example, in figure-1, the construction of the hierarchical projection network appears erratic with the various skip connections and the curved lines, what do they represent in the figure?
b) Figure-2 appears chaotic, flow of arrows into different parts of the network are not clear, and the patch step appears unorganized, making it difficult to understand what are the queries and what exactly are the key-values in the network. The relation between the upsampling and downsampling layers are not also clear, do they act as residual connections? How are the attention blocks and upsampling blocks arranged with respect to each other?"

---

> ### Author Rebuttal · Authors · 2026-03-31
>
> Dear Reviewer,
>
> W1:The target problem (Lines 10-18) and the proposed solution (Lines 22-32) are already stated in the abstract, and we will further polish this part for clarity. As introduced in Secs. 3.2-3.3, spectral prompts bridge the coordinate and latent spaces, upon which the trigonometric basis solver performs reconstruction as a downstream module.
>
> W2:a) The Latent-token Cross-attention Projectors map coordinate-space features into a latent space, where the main fusion is performed independently of the dense coordinate grid. Thus, the decoupling refers to the fusion process, rather than to the whole network.
>
> b) The references to large language models and PDE models are included to motivate the potential of latent-space learning. Our intention is to highlight the broader effectiveness of performing learning and inference in latent representations. Following your suggestion, we will reduce this type of references and add more references on reconstruction tasks.
>
> c) The paragraph beginning with “Based on this insight” motivates the design of LSO, while the bullet list summarizes the formal contributions. We will further polish these two parts to make them more distinct.
>
> W3:The Introduction briefly presents SSF and latent-space learning, while the Related Work expands on them in greater detail to provide the necessary background and context for our method. We will also consider your suggestion and try to merge them together.
>
> W4:a) We will consider refining the title and further improving the description of the overall architecture (Lines 121-142, left).
>
> b)&Q2:We respectfully clarify that the input channels correspond to sampled spectral bands and channel mixing in the convolutional encoder should not be interpreted as eliminating all spectral information. EDSR is adapted to the HR-MSI and upsampled LR-HSI inputs to extract joint spatial–spectral features, while the subsequent Hierarchical Projection Network and trigonometric-basis solver explicitly model cross-band dependencies in latent space. Thus, “spectral information” in our method refers to structured latent spectral modeling rather than preservation of raw unmixed bands.
>
> The statement refers not to channel mixing itself, but to unconstrained latent fusion, which may arbitrarily entangle cross-band relationships and cause inconsistent spectral responses in the reconstructed HR-HSI. We will revise the wording for clarity.
>
> c)&Q3:After the Latent-token Cross-attention Projectors map EDSR features into the latent space, the Trigonometric Basis Solver learns and fuses them as formulated in Eq. (7). Instead of undoing the encoder’s channel mixing, the solver imposes an explicit basis-structured parameterization on latent spectral representations to regularize cross-band variation, reduce unstructured entanglement and improve spectral consistency.
>
> W5:a) In fact, the dataset construction process is already described in the main text (Lines 248-263). Specifically, benchmark HR-HSIs are treated as pseudo ground truth, and the LR-HSI/HR-MSI observations are synthetically generated using the prescribed degradation operators. This is a widely adopted practice in SSF literature [Otias, SpecSolver, KNLConv, FeINFN], rather than a special assumption introduced by our method.
>
> b)&Q5:As shown in https://anonymous.4open.science/r/6909_rebuttal_Visualization-C99D/README.md, we visualize the representations at different stages of the pipeline. EDSR serves as the front-end encoder, extracting local spatial details and initial joint spatial–spectral features from the input observations. The Hierarchical Projection Network is then introduced to project features on the dense coordinate grid into a compact latent space, where the model can organize multi-level information and perform feature fusion more effectively than in the original coordinate space. The Reconstruction Head adjusts the channel dimension so that the output matches the number of channels in the ground-truth HR-HSI.
>
> W6&Q4:Figures 1 and 2 illustrate the overall pipeline rather than operator-level implementations. In Fig. 1, the skip connections denote residual fusion and the curved lines indicate cross-scale information flow rather than extra operations. In Fig. 2, the definitions of queries, keys, and values are given in Sec. 3.2 and Eqs. (2)(3), while the details of the upsampling and related modules are provided in Appendix A. The downsampled multi-scale features are processed by different Hierarchical Projection Networks with attention modules and then upsampled for reconstruction. We will improve the captions and annotations to make these relations clearer.
>
> Q1:The Latent-token Cross-attention Projectors are not designed for geometric alignment, since the two modalities are already pixel-aligned. They project the input features onto shared latent tokens to aggregate full spatial information into a compact latent representation (Lines 110-129, right).
>
> **Thanks for your valuable comments!**

---

> > ### Author Rebuttal · Reviewer_jWoN · 2026-04-04
> >
> > Thank you to the authors for the rebuttal. The responses have addressed some of my concerns. I would like to stay with my current rating of the paper.

---

### Official Review · Reviewer_TxwP · 2026-03-17

**Soundness:** 3
**Presentation:** 4
**Significance:** 3
**Originality:** 3
**Overall Recommendation:** 4
**Confidence:** 4

**Summary:**

The paper proposes a novel Spatial-Spectral Fusion framework named Latent Spectral Operators (LSO). Unlike traditional methods that learn fusion mappings directly in the coordinate domain using convolutions or attentions, LSO projects spatial features into a coordinate-free latent space via cross-attention and modulates spectral features using a Trigonometric-Basis Solver. This methodology is conceptually designed to tackle the performance degradation typical in cross-resolution generalization, and to explicitly control high-frequency injection to prevent spectral distortion. The authors provide extensive empirical evaluations, and notably, a highly rigorous set of ablation studies and statistical significance tests in the Appendix, demonstrating an excellent level of engineering completeness.

**Compliance With Llm Reviewing Policy:**

Affirmed.

**Key Questions For Authors:**

How does the EDSR spatial encoder maintain semantic consistency in its feature maps when tested on extreme scale factors (e.g., x16 or x32), given its fixed convolutional receptive field? Can the authors visualize or quantify the robustness of these pre-latent convolutional features across scales?

**Limitations:**

Yes.

**Strengths And Weaknesses:**

Strengths:
1. The paper introduces a highly inspiring paradigm shift by transitioning the SSF task from traditional "coordinate-domain pixel mapping" to "latent-space equation solving." This novel approach offers a valuable new perspective for breaking the resolution dependency that is inherently problematic in many existing vision models.
2. The proposed Trigonometric-Basis Solver provides an elegant mathematical foundation for explicitly controlling high-frequency information injection. Furthermore, the Fourier amplitude analysis provided in the appendix convincingly supports why this design successfully avoids the "spiky peaks" common in standard self-attention mechanisms.
3. The framework demonstrates exceptional empirical robustness, supported by strong zero-shot scaling performance on benchmarks and highly rigorous ablation studies. The comprehensive tests provided in the appendix, which confirm the model's stability regarding critical hyper-parameters like the number of tokens and patch sizes, strongly solidify the empirical reliability of the proposed method.

Weakness:
1. Potential theoretical contradiction between the scale-variant spatial encoder and the "resolution independence" claim. While LSO elegantly achieves resolution decoupling in the latent space, its initial feature extraction heavily relies on EDSR, a standard CNN with a strictly fixed receptive field. At extreme test scales (e.g., x16 or x32), the physical semantic region covered by this fixed receptive field changes drastically compared to the training phase (e.g., x4). It seems theoretically contradictory to claim pure resolution independence when the preliminary feature maps fed into the LSO module are inherently scale-variant. Although the empirical metrics at extreme scales are impressive, it would be highly convincing if the authors could provide a visual or statistical comparison of the EDSR feature maps across different scale factors in the rebuttal. I would like to understand how these initial semantic features maintain consistency under extreme scaling, or how the LSO module corrects this inherent CNN scale-variance.

---

> ### Author Rebuttal · Authors · 2026-03-31
>
> # Response to the Reviewer TxwP
> We sincerely thank the reviewer for recognizing the **novelty** and **reliability** of our manuscript and for providing valuable and constructive suggestions. Our responses are as follows.
>
> We agree that the EDSR front-end is not strictly resolution-invariant. Due to its fixed convolutional receptive field, the pre-latent features exhibit visible scale-dependent drift, especially at extreme factors such as ×32. Our claim is therefore not strict resolution independence of the spatial encoder itself, but reduced resolution dependence achieved by the downstream latent projection and latent-space solver. Although the pre-latent CNN features are scale-variant, they still preserve coarse structural cues that remain alignable across scales.
>
> In the rebuttal (https://anonymous.4open.science/r/6909_rebuttal_Visualization-C99D/README.md), we provide stage-wise visualizations across scales (EDSR Features → Latent Features after CoordToLatent → Latent Features after Trigonometric-Basis Solving → Decoded Spatial Features), which show that cross-scale drift is progressively reduced throughout the pipeline. Quantitatively, as shown in Table 1, using ×4 as the reference scale, the mean cosine similarity consistently increases from the pre-latent EDSR feature space to the latent-projected space. Moreover, the cosine similarity is further improved after solving with the trigonometric basis. However, although the similarity improves, reconstruction quality at larger scaling factors (e.g., ×32) still deteriorates due to the loss of fine details in the initial stage. This also explains the slight performance drop observed in Fig. 7 of the manuscript as the scaling factor increases.
>
> This confirms that our method does not assume strict scale invariance of the front-end CNN, but progressively reduces scale-induced drift through the latent projection and latent-space solver. This interpretation is also consistent with prior operator-based super-resolution frameworks such as Super-Resolution Neural Operator, where EDSR is better understood as a local feature extractor or common basis initializer, while the downstream coordinate-aware latent/operator module is responsible for alleviating scale or discretization dependence and enabling cross-resolution generalization.
>
> **We will incorporate the above experimental results and analysis into the camera-ready version to make this distinction explicit. Thank you once again for your highly constructive suggestions!**
>
> ---
>
> **Table 1.** Cross-scale cosine similarity at different stages of the pipeline (taking ×4 as the reference; higher is better).
>
> | Stage | ×4 vs ×8 | ×4 vs ×16 | ×4 vs ×32 |
> |---|---:|---:|---:|
> | EDSR Features | 0.824 ± 0.027 | 0.571 ± 0.038 | 0.395 ± 0.043 |
> | Latent Features after CoordToLatent | 0.889 ± 0.021 | 0.697 ± 0.033 | 0.533 ± 0.039 |
> | Latent Features after Trigonometric-Basis Solving | 0.915 ± 0.017 | 0.776 ± 0.029 | 0.638 ± 0.034 |
> | Decoded Spatial Features | 0.891 ± 0.020 | 0.732 ± 0.031 | 0.586 ± 0.036 |

---

### Decision · Program_Chairs · 2026-04-30

**Decision:**

Accept (regular)

**Comment:**

Latent Spectral Operators (LSO) are proposed, for spatial–spectral fusion. Operators work on latent space, while deviating from typical  coordinate domain fusion. The reviewers more or less agree that the paper addresses an important problem, is technically sound, and that it also presents strong results, in particular when it comes to cross-resolution robustness. Several reviewers liked the originality of the latent operator approach, the effectiveness of the trigonometric solver, and the thorough empirical experimentation, with due ablations.
The 'negative review' took mainly issue with aspects like the paper's clarity and presentation style. Moreover, it raised concerns about whether spectral information is preserved through the encoder and whether the role of several of the modules is clear enough. These concerns are not trivial, and I agree that the paper could be improved based on these comments, as the rebuttal already started to do. However, I do not see evidence of a fundamental technical flaw. The authors’ rebuttal addressed a number of the important misunderstandings. While one reviewer remained unconvinced, all other reviews lean towards acceptance. Overall, I believe the paper should be accepted. There is enough novelty and the method is sufficiently solid. The camera-ready version will hopefully come with improved clarity, will best also discuss limitations, and include the changes promised in the rebuttal.